# Mental health in autistic adults: A rapid review of prevalence of psychiatric disorders and umbrella review of the effectiveness of interventions within a neurodiversity informed perspective

Eleanor Curnow[1]*, Marion Rutherford[1], Donald Maciver[1], Lorna Johnston[1,2], Susan Prior[1], Marie Boilson[1,3], Premal Shah[1,4], Natalie Jenkins[1,5], Tamsin Meff[1]

1 School of Health Sciences, Queen Margaret University, Edinburgh, United Kingdom, 2 Additional Support for Learning Service, Communities and Families, City of Edinburgh Council, Edinburgh, United Kingdom, 3 Fife Health and Social Care Partnership, Lynebank Hospital, Dunfermline, Fife, United Kingdom, 4 General Adult Psychiatry, Royal Edinburgh Hospital, Edinburgh, United Kingdom, 5 University of Edinburgh, Edinburgh, United Kingdom

* ecurnow@qmu.ac.uk

## Abstract

### Background

Autistic adults have high risk of mental ill-health and some available interventions have been associated with increased psychiatric diagnoses. Understanding prevalence of psychiatric diagnoses is important to inform the development of individualised treatment and support for autistic adults which have been identified as a research priority by the autistic community. Interventions require to be evaluated both in terms of effectiveness and regarding their acceptability to the autistic community.

### Objective

This rapid review identified the prevalence of psychiatric disorders in autistic adults, then systematic reviews of interventions aimed at supporting autistic adults were examined. A rapid review of prevalence studies was completed concurrently with an umbrella review of interventions. Preferred Reporting Items for Systematic Review and Meta-Analysis (PRISMA) guidelines were followed, including protocol registration (PROSPERO#CRD42021283570).

### Data sources

MEDLINE, CINAHL, PsycINFO, and Cochrane Database of Systematic Reviews.

### Study eligibility criteria

English language; published 2011–2022; primary studies describing prevalence of psychiatric conditions in autistic adults; or systematic reviews evaluating interventions for autistic adults.

**Data Availability Statement:** All relevant data are within the paper and its supporting information files

**Funding:** This study was supported by funding from Scottish Government. The funders had no role in study design, data collection and analysis, decision to publish, or preparation of the manuscript.

**Competing interests:** The authors have declared that no competing interests exist.

## Appraisal and synthesis

Bias was assessed using the Prevalence Critical Appraisal Instrument and AMSTAR2. Prevalence was grouped according to psychiatric diagnosis. Interventions were grouped into pharmacological, employment, psychological or mixed therapies. Strength of evidence for interventions was assessed using GRADE (Grading of Recommendations, Assessment, Development and Evaluation). Autistic researchers within the team supported interpretation.

## Results

Twenty prevalence studies were identified. Many included small sample sizes or failed to compare their sample group with the general population reducing validity. Prevalence of psychiatric diagnoses was variable with prevalence of any psychiatric diagnosis ranging from 15.4% to 79%. Heterogeneity was associated with age, diagnosis method, sampling methods, and country. Thirty-two systematic reviews of interventions were identified. Four reviews were high quality, four were moderate, five were low and nineteen critically low, indicating bias. Following synthesis, no intervention was rated as 'evidence based.' Acceptability of interventions to autistic adults and priorities of autistic adults were often not considered.

## Conclusions

There is some understanding of the scope of mental ill-health in autism, but interventions are not tailored to the needs of autistic adults, not evidence based, and may focus on promoting neurotypical behaviours rather than the priorities of autistic people.

## Introduction

Mental ill-health is a common experience for autistic adults [1]. The recent Lancet Commission on autism research described a 'deep scarcity' of evidence regarding interventions and supports for this population [2, 3]. Considering the recent increase in interest in outcomes for autistic adults, research and policy advancements in this field are urgently required.

There is a known increased risk for experiencing mental ill-health in autism this varies widely in terms of reported prevalence [3]. Estimated prevalence of autism in adults aged 16–64 years in UK is 2.9% [95% CI 2.7, 3.1] [4]. Prevalence of autism is 3.46 times higher for boys [5]. Autistic people have a wide range of needs which vary depending on environment, and co-occurrence of intellectual or physical factors, sensory factors, co-occurring neurodevelopmental differences, intellectual disabilities, or other psychiatric diagnoses [6–8]. Autistic people, and people with intellectual disabilities have more mental and physical needs than other people [9], and research indicates that needs prevalence will be even higher for people with co-occurring autism and intellectual disability [8].

Worldwide prevalence of psychiatric disorders is estimated at 13%, including anxiety disorders (4.1%), depressive disorders (3.8%), bipolar disorders (0.5%), schizophrenia (0.3%), and eating disorders (0.2%) [10]. In Scotland, census data indicates that 5.4% of adults aged 16–64 years (4.6% for people aged 65+) without co-occurring intellectual disabilities and autism reported mental ill-health which had lasted or was expected to last at least 12 months [8].

Whilst there has been previous consideration of prevalence of psychiatric disorders in autistic populations [11], there is a need to distinguish between adult and child populations. Further consideration of the measurement tools used with autistic adults is also required to ensure that they are validated for this population [12]. The significance of co-occurring mental ill-health was identified in research which demonstrated that that up to 66% of autistic adults without intellectual disability have contemplated suicide compared to 17% of non-autistic adults, and research has linked this to social camouflaging [13].

Mental health has been identified as a top priority research area for autistic adults [14]. Mental health is a state of well-being in which an individual realises his or her own abilities, can cope with the normal stresses of life, can work productively and is able to make a contribution to his or her community [15]. As understanding of mental health in autism evolves, it is also recognised that personal factors cannot be separated from environmental factors. Attitudes, understanding and expectations of those around an individual and adaptations in society and everyday environments are fundamental to supporting meaningful participation and positive mental health in autism [16].

For the purposes of this review a psychiatric disorder is defined as a mental illness diagnosed by a mental health professional according to diagnostic criteria [17]. Relevant diagnoses were identified according to search terms and strategies described by Cochrane Common Mental Disorders [18].

There is limited understanding of effective interventions for supporting mental health in autistic adults [2]. A recent umbrella review found that research evidence did not support one best intervention for autism in children, and that there was a concerning lack of consideration of adverse effects of interventions [19]. Previous research has focussed on children and adolescents, often evaluating interventions designed to reduce or mask behaviours associated with autism [20] but there is now recognition of the stress and detriment such interventions can create [20, 21]. The 'neurodiversity' movement considers autism and other neurodevelopmental conditions as neurological variation, rather than disorders requiring treatment [2, 22, 23], Therefore, autism is a difference not a deficit, which brings into question the use of interventions which seek to 'cure, fix or normalise' [2]. This movement has provided tools to critique research and to consider what is important in research and practice for autistic adults [16, 23, 24]. This has led to the development of research priorities which focus on the best interests of autistic people and recognise that the inclusion of both autistic people and non-autistic people in research processes is of key importance [22]. Although, there is a need for progress as only 5% of funded autism research included autistic adults [25]. Historical research must be reviewed through a contemporary lens which considers the acceptability of terminology, interventions, supports and outcomes to the autistic community [23]. Research indicates that autistic people prioritise outcomes associated with quality of life, reduction in anxiety, depression or sleep related problems, social well-being, interpersonal relationships, and increased participation in activities of daily living, community, and work [24].

These measures are key to evidence-based practice which requires the integration of the best available research with clinical expertise and the patient's unique values and circumstances [26, 27]. Evidence based practice requires that health care is not only based upon the best available, valid, and current evidence as defined by GRADE (Grading of Recommendations, Assessment, Development and Evaluation) [28], but also that decisions are made by those receiving care and informed by those providing care [27, 29]. Strong GRADE evidence indicates all or almost all people would choose that intervention [30]. This umbrella review of interventions will therefore consider the results of studies not only in terms of their effectiveness, but also regarding the acceptability of the interventions to the autistic community [23].

The prevalence of psychiatric disorders in autistic adults will be explored through rapid review of published literature. This knowledge synthesis will be rigorous and transparent but will be accelerated by resource-efficient methods including limiting the number of databases which will be searched for evidence. Handsearching, and forward and backward citation searches will also not be undertaken [31]. Grey literature, and literature not published in English will not be considered. Article screening will be reviewed by two authors in 20% of publications.

An umbrella review facilitates a synthesis and appraisal of evidence across a broader topic area than can usually be achieved through an individual systematic review [32]. In the current research, the aim was to incorporate these key viewpoints, integrating perspectives on evidence-based practice, and views from people with lived experience, experts, and practitioners. These key ideas are summarised below.

## Objectives

We conducted a rapid review of existing studies providing quantitative data on the prevalence of psychiatric diagnoses in autistic adults. We also conducted an umbrella review of systematic reviews of interventions for autistic adults [32]. In both cases following Preferred Reporting Items for Systematic review and Meta-analyses (PRISMA) guidelines [33] (S1 Checklist). Our research was commissioned to inform an adult autism government policy review in Scotland which aimed to set priorities which are driven by autistic people. Considering these principles, our objectives for this review focussed on autistic adults were to:

1. Establish prevalence of psychiatric diagnoses and explore associated heterogeneity.

2. Investigate evidence for effectiveness of interventions.

3. Consider the acceptability of interventions and research in this field with reference to the neurodiversity paradigm.

### Research questions.

1. How prevalent are psychiatric diagnoses in autistic adults?

2. Which factors are associated with heterogeneity of prevalence of psychiatric diagnoses in autistic adults?

3. Which interventions are effective in treating autistic adults?

4. Do available interventions meet the needs and priorities of autistic adults?

## Methods

The systematic review process was undertaken in two parts focussing on (a) primary prevalence data describing the occurrence of psychiatric diagnoses in autistic adults; and (b) umbrella reviews of interventions. This was a rapid review, and to reduce the time required limited databases were included in the searches, and the umbrella review of interventions considered systematic reviews only, due to their higher quality research design [34, 35]. Search date was restricted to 10 years as this is a valid and reliable approach for rapid reviews [36]. The protocol was registered a priori (PROSPERO #CRD42021283570).

### Inclusion criteria

For both reviews, studies were included if: a) participants were autistic (however defined in the study which may include self-diagnosis or clinical diagnosis) b) participants were ≥18 years of age (or the mean age of the participant group ≥18 years) c) they were reported in English; d) they were published from 01/2011,

 AND

1. For the investigation of prevalence of psychiatric diagnoses in autistic adults; studies were included if they reported primary prevalence data for occurrence of psychiatric diagnoses experienced by autistic adults.
   OR

2. For the umbrella review of interventions; studies were included if a) they considered interventions for autistic adults; b) they were systematic reviews.

### Search strategy

A systematic search of MEDLINE, CINAHL (Cumulative Index to Nursing and Allied Health Literature) and PsycINFO databases was conducted in November 2022, through EBSCOhost to identify quantitative studies of psychiatric diagnoses in autistic adults using Medical Subject Headings (MeSH) and keywords. To identify systematic reviews of interventions, a systematic search of CINAHL, MEDLINE, PsycINFO, and Cochrane Database of Systematic Reviews was conducted in November 2022.

 Databases were selected from available resources following current guidance [37], and through discussion with the university research librarian. CENTRAL, MEDLINE and Embase (if access to Embase is available to the review team) are recommended for systematic reviews [37–39]. Embase was not available to the research team. Cochrane Database of Systematic Reviews was included as a major repository of systematic reviews [32]. Trials searches of JBI Database of Systematic Reviews and Implementation Reports did not reveal any additional relevant citations and was therefore excluded. Lists of search terms are included (S1 File).

### Study selection

Retrieved citations were uploaded to Covidence [40]. Following removal of duplicates, titles and abstracts of the returned articles were examined blind by two researchers (EC, NJ) with irrelevant titles excluded. Full text articles were then reviewed against inclusion criteria (EC), with 20% examined by a second reviewer (NJ). Disagreements were resolved through discussion, and reference to a third party (MR) was not required. Inter-rater agreement was assessed using Cohens Kappa.

### Data extraction

Data extraction sheets were developed and piloted by two researchers. For prevalence, extracted data included study details, setting, sample size, age, method of diagnosis, prevalence data for: any psychiatric diagnosis; anxiety; depression; psychosis; schizophrenia; obsessive-compulsive disorder; attention deficit hyperactivity disorder (ADHD); bipolar disorder; eating disorder. For interventions, extracted data included study details, methodology, aims, population age and gender, context, inclusion/exclusion criteria, intervention, psychiatric diagnosis, autism diagnosis, sample size, subgroups, views of autistic adults, concerns re acceptability of intervention, conclusions, recommendations, funding sources. Data extraction was conducted

by one of the research team members, then reviewed by the research team as a group, inconsistencies were resolved through discussion.

## Evaluation of risk of bias

Studies included in the prevalence review were assessed for risk of bias using the Prevalence Critical Appraisal Instrument [41]. This assessment focusses on a) method of identification or diagnosis of the relevant condition, and b) sampling of the population, as these are issues particularly relevant to prevalence. Studies included in the intervention review were assessed for risk of bias using AMSTAR2 [42]. This tool identifies domains critical to integrity of the study, including registration of protocol (Q2), adequacy of literature search (Q4), justification for excluding studies (Q7), risk of bias in inclusion of studies (Q9), selection of meta-analysis methods (Q11), consideration of risk of bias in interpretation of results (Q13), consideration of publication bias (Q15). Studies are rated as high, moderate, low, or critically low according to the number of weaknesses identified [42]. Risk of bias was assessed by one member of the research team, then reviewed by another.

## Synthesis methods

For synthesis of prevalence, studies were pooled according to psychiatric diagnosis.

For synthesis of interventions, results were grouped into pharmacological interventions; employment focussed interventions; psychological therapies; and mixed intervention or approaches. Within these broad categories, a list of detailed intervention sub-categories was identified. Next, evaluation of the strength of evidence for each sub-category was completed based upon the following criteria adapted from GRADE [28] from 'not recommended- to 'evidence based' (Table 1).

Interventions were evaluated against the stated adapted GRADE criteria to determine not only evidence of effectiveness, but also evidence of negative consequences or harm. This involved consideration of reported benefit for each intervention type. An exploratory approach was used to review adverse outcomes identified during the conduct of the review. This opportunistic approach considers only the reported adverse effects or outcomes that may be associated with the interventions being investigated [43].

In considering negative consequences or harm associated with interventions we included criteria adapted from clinical guidelines and neurodiversity affirming practice. Specifically, we did not recommend:

- Interventions which focussed on the reduction of core features of autism are associated with harmful consequences and contradict current clinical guidelines [44, 45]. Core features

**Table 1. Research recommendation classification.**

| Level of evidence | Description |
|---|---|
| Not recommended | There is no evidence of effectiveness, or there is evidence of negative consequences or harm |
| Unestablished evidence | All studies showed no effect or there was only one study with that intervention available for review |
| Emerging evidence | Two or more RCTs of lower quality, less rigorous study design and available evidence showed some or no effect, with no negative effects |
| Evidence based | Two or more high quality RCTs to support the intervention or five high quality single subject design studies conducted by at least three different research groups |

RCT = randomised controlled trial

include qualitative differences and impairments in reciprocal social interaction and social communication, restricted interests and activities, and rigid and repetitive behaviours [46].

- Interventions which contradicted current clinical guidelines [44, 46].

- Interventions associated with adverse events or adverse outcomes [43].

- Interventions which attempt to 'cure, fix or normalise' autistic people [2, 47] due to their negative impact upon quality of life [29].

- Interventions which target outcomes contradictory to the identified priorities of the autistic community [14, 24, 48–50].

The research team was made up of autistic and non-autistic professionals within speech and language therapy, psychology, psychiatry, occupational therapy, and teaching fields. Members of the team had research experience, and experience working with autistic people in clinical and education settings. As integrated members of the research team, autistic researchers contributed to the planning and design of this research study, and decision-making related to study outcomes alongside non-autistic colleagues. All team members held professional roles and contributed expertise to the study thus possibly reducing issues associated with power hierarchy sometimes found in autism research [51]. Arising disagreements concerning the classification of evidence were resolved through team discussion with reference to research recommendation classification (Table 1) and criteria regarding negative consequences or harm listed above until agreement was achieved. Inter-rater reliability was not recorded for this process.

## Results

The database searches returned 283 papers for the prevalence review and 448 papers for the interventions review. Of these 20 papers describing the prevalence of psychiatric diagnoses for autistic adults (Fig 1), and 32 papers describing interventions for autistic adults (Fig 2), met the inclusion criteria. Citations for excluded papers are provided (S2 File).

Cohen's kappa of inter-rater reliability was 0.70 for both studies, indicating substantial agreement between reviewers.

### Prevalence

Characteristics of included papers relating to prevalence are described (Table 2), together with a summary of issues highlighted during completion of the Prevalence Critical Appraisal Instrument [41]. Studies often included small sample sizes or failed to compare their sample group with the general population reducing understanding of the validity of their findings. Prevalence data was not pooled due to heterogeneity [52], which was associated with age, co-occurring conditions, sampling method, mode of diagnosis, variation in the categorisation or grouping of diagnoses, and the country in which the study took place. Prevalence ranges, across all studies, locations, and lifetime vs current diagnosis, were any psychiatric diagnosis 15%-79%; Attention-deficit hyperactivity disorder 2%-33%; Depression 10%-54%; Anxiety 10%-54%; Psychotic disorders 0.2%-18%; bipolar disorder 1%-25%; obsessive compulsive disorder 2%-33%; and eating disorders 2%-11% (Table 2).

### Interventions

Characteristics of the systematic reviews relating to interventions for autistic adults are described in Table 3. This includes effect size from any data synthesis conducted within the

**Table 2. Prevalence study characteristics.**

| Study | Setting | Sample size (N) | Gender %Male | Age (years) | ID Diagnosis % | Psychiatric Diagnosis | Lifetime (L)/ Current (C) | Prevalence | Summary of Issues which may impact prevalence |
|---|---|---|---|---|---|---|---|---|---|
| [53] | Outpatient Psychiatric Unit, Sweden | 50 | 52% | 20–47 | 0 | Anxiety | C | 0.28 | Small clinical sample, characteristics not compared with general population |
| [54] | Population based study, Utah | 129 | 75.10% | 26–54, M = 36.4 (SD = 5.9) | | Any Diagnosis | C | 0.57 | No comparison, some subjects diagnosed from clinical records alone. |
| | | | | | | | L | 0.69 | |
| | | | | | | Anxiety | C | 0.4 | |
| | | | | | 76.7 | | L | 0.53 | |
| | | | | | | Depression | C | 0.12 | |
| | | | | | | | L | 0.13 | |
| | | | | | | Psychosis | C | 0.05 | |
| | | | | | | | L | 0.1 | |
| | | | | | | OCD | C | 0.33 | |
| | | | | | | | L | 0.36 | |
| [55] | Kaiser Permanente Northern California Medical Records | 4123 | 80.67% | 14–25, M = 18.4 (SD = 3.2) | 12% (18–21 years) 19% (22–25 years) | Any Diagnosis | L | 0.34 | Large sample group compared with other clinical groups and typical control groups. |
| | | | | | | Anxiety | | 0.14 | |
| | | | | | | Depression | | 0.1 | |
| | | | | | | Psychosis | | 0.02 | |
| | | | | | | OCD | | 0.02 | |
| | | | | | | ADHD | | 0.15 | |
| | | | | | | Bipolar Disorder | | 0.06 | |
| [56] | Neuropsychiatric clinic, Milan | 106 | 73% | 17–67, M = 18.4 (SD = 12.88) | 0 | Any Diagnosis | C | 0.25 | Clinical sample compared with neurotypical adults. |
| [8] | Census Data, Scotland | 3103 | 66.80% | 16–64 | 100 | Any diagnosis | L | 0.37 | Large sample of autistic people with co-occurring intellectual disabilities, census completed by 94% population, self-report of diagnoses |
| | | 244 | 83.12% | 65+ | 100 | | | 0.66 | |
| [57] | Neurodevelopmental Disorders Lab, South Carolina | 20 | 100% | 13–22, M = 18.94 (SD = 2.20) | Brief IQ: M = 68.15 (SD = 26.2) | Anxiety | C | 0.5 | Small clinical sample, compared with FXS population. |
| [58] | Disease Register/ Electronic Health Record, New York, USA | 116 | 75.30% | 18–29 (M = 24.1) | | Anxiety | L | 0.35 | Clinical sample- compared with general population. Limited data on IQ of sample. |
| | | | | | 13%—normal IQ, No IQ data for 45% of sample. | Depression | | 0.16 | |
| | | | | | | Psychosis | | 0.04 | |
| | | | | | | ADHD | | 0.28 | |
| | | | | | | Bipolar Disorder | | 0.08 | |
| | | 67 | | 30–39 (M = 33.8) | | Anxiety | L | 0.22 | |
| | | | | | | Depression | | 0.18 | |
| | | | | | | Psychosis | | 0.03 | |
| | | | | | | ADHD | | 0.09 | |
| | | | | | | Bipolar Disorder | | 0.03 | |
| | | 72 | | 40–71 (M = 48.8) | | Anxiety | L | 0.32 | |
| | | | | | | Depression | | 0.1 | |
| | | | | | | Psychosis | | 0.04 | |
| | | | | | | ADHD | | 0.03 | |
| | | | | | | Bipolar Disorder | | 0.04 | |

(*Continued*)

**Table 2.** (Continued)

| Study | Setting | Sample size (N) | Gender %Male | Age (years) | ID Diagnosis % | Psychiatric Diagnosis | Lifetime (L)/ Current (C) | Prevalence | Summary of Issues which may impact prevalence |
|---|---|---|---|---|---|---|---|---|---|
| [59] | Medicare Data, USA | 4685 | 67.80% | 65+ | | Anxiety | L | 0.37 | Large sample, compared with matched population. Older population often diagnosed prior to DSM criteria. |
| | | | | | | Depression | | 0.36 | |
| | | | | | 43.8 | Psychosis | | 0.18 | |
| | | | | | | ADHD | | 0.02 | |
| [60] | Systematic Review | 26070 | NA | M = 30.9 (SD = 6.2); | | Anxiety | C | 0.27 | Not empirical study. Subgroup analysis investigated impact of intellectual disability and method of diagnosis. Includes data from other studies included in this review. |
| | | | | | | | L | 0.42 | |
| | | 26117 | | M = 31.1 (SD = 6.8) | NA | Depression | C | 0.19 | |
| | | | | | | | L | 0.4 | |
| | | NA | | NA | | OCD | C | 0.22 | |
| | | | | | | | L | 0.24 | |
| [61] | USA Commercial | 8370 | | 18–24 | | Anxiety | L | 0.32 | Insurance database, compared with Medicaid data. Participants may be included in both Medicaid and Commercial Database. |
| | | | | | | Depression | | 0.2 | |
| | | | | | 8.43 | Psychosis | | 0.03 | |
| | | | | | | ADHD | | 0.33 | |
| | | | | | | Bipolar Disorder | | 0.1 | |
| | | 2722 | | 25–49 | | Anxiety | | 0.31 | |
| | | | | | | Depression | | 0.21 | |
| | | | | | 14.92 | Psychosis | | 0.04 | |
| | | | | | | ADHD | | 0.18 | |
| | | | | | | Bipolar Disorder | | 0.1 | |
| | | 386 | | 50+ | | Anxiety | | 0.36 | |
| | | | | | | Depression | | 0.3 | |
| | | | | | | Psychosis | | 0.04 | |
| | | | | | 9.07 | | | | |
| | | | | | | ADHD | | 0.17 | |
| | | | | | | Bipolar Disorder | | 0.1 | |
| | USA Medicaid | 6716 | | 18–24 | | Anxiety | L | 0.18 | Insurance database, compared with Commercial data. Participants may be included in both Medicaid and Commercial Database. |
| | | | | | | Depression | | 0.15 | |
| | | | | | 34.86 | Psychosis | | 0.06 | |
| | | | | | | ADHD | | 0.3 | |
| | | | | | | Bipolar Disorder | | 0.15 | |
| | | 3807 | | 25–49 | | Anxiety | | 0.17 | |
| | | | | | | Depression | | 0.14 | |
| | | | | | 57.81 | Psychosis | | 0.1 | |
| | | | | | | ADHD | | 0.11 | |
| | | | | | | Bipolar Disorder | | 0.13 | |
| | | 252 | | 50+ | | Anxiety | | 0.24 | |
| | | | | | | Depression | | 0.24 | |
| | | | | | 64.68 | Psychosis | | 0.17 | |
| | | | | | | ADHD | | 0.02 | |
| | | | | | | Bipolar Disorder | | 0.15 | |

(*Continued*)

**Table 2.** (Continued)

| Study | Setting | Sample size (N) | Gender %Male | Age (years) | ID Diagnosis % | Psychiatric Diagnosis | Lifetime (L)/ Current (C) | Prevalence | Summary of Issues which may impact prevalence |
|---|---|---|---|---|---|---|---|---|---|
| [62] | CPRD Data, UK | 2467 | 80.70% | 18–24 | | Anxiety | L | 0.1 | Large sample, compared with general population and ADHD groups. Previous research found Autism diagnoses in CPRD to be reliable. |
| | | | | | | Depression | | 0.1 | |
| | | | | | | Psychosis | | 0.002 | |
| | | | | | 1.7 | OCD | | 0.03 | |
| | | | | | | ADHD | | 0.17 | |
| | | | | | | Bipolar Disorder | | 0.03 | |
| | | 1667 | | 25–49 | | Anxiety | | 0.18 | |
| | | | | | | Depression | | 0.28 | |
| | | | | | | Psychosis | | 0.02 | |
| | | | | | 13.9 | OCD | | 0.06 | |
| | | | | | | ADHD | | 0.1 | |
| | | | | | | Bipolar Disorder | | 0.02 | |
| | | 428 | | 50+ | | Anxiety | | 0.19 | |
| | | | | | | Depression | | 0.35 | |
| | | | | | | Psychosis | | 0.05 | |
| | | | | | 26.2 | OCD | | 0.05 | |
| | | | | | | ADHD | | 0.01 | |
| | | | | | | Bipolar Disorder | | 0.06 | |
| [63] | Systematic Review | NA | | 18+ | NA | Depression | C | 0.19 | Not empirical study. Small number of studies in meta-analysis. |
| | | | | | | | L | 0.4 | |
| [64] | Neuropsychiatric Genetic Study, Gothenburg | 74 (Autism) | 55.70% | 19–57, M = 31.75 (SD = 9.29) | NA | Eating Disorder | C | 0.11 | Clinical sample. Autism compared with Autism+ ADHD and ADHD groups. Diagnosis of eating disorder made using single diagnostic tool. |
| | | 45 (Autism + ADHD) | | | | | | 0.02 | |
| [65] | Mental Health Institutions, Client organisations; Netherlands | 138 | NA | 19–79, M = 46.5 | | Any diagnosis | C | 0.79 | Clinical sample with comparison group. |
| | | | | | | Anxiety | | 0.54 | |
| | | | | | IQ>80 | Depression | | 0.54 | |
| | | | | | | OCD | | 0.3 | |
| | | | | | | ADHD | | 0.3 | |
| | | | | | | Eating Disorder | | 0.05 | |
| [66] | Neurodevelopmental outpatient clinic, Athens | 58 | 81% | M = 28.7 (SD = 9.2) | | Anxiety | C | 0.14 | Small clinical sample, Autistic Group |
| | | | | | | Depression | | 0.29 | |
| | | | | | IQ>70 | Psychosis | | 0.07 | |
| | | | | | | OCD | | 0.09 | |
| | | | | | | Bipolar Disorder | | 0.03 | |
| | | 29 | 65.50% | M = 28.8 (SD = 10) | | Anxiety | C | 0.1 | Small clinical sample, ADHD+ Autism Group |
| | | | | | | Depression | | 0.24 | |
| | | | | | | Psychosis | | 0.1 | |
| | | | | | IQ>70 | OCD | | 0.24 | |
| | | | | | | Bipolar Disorder | | 0.14 | |

*(Continued)*

**Table 2.** (Continued)

| | Study | Setting | Sample size (N) | Gender % Male | Age (years) | ID Diagnosis % | | | Psychiatric Diagnosis |
|---|---|---|---|---|---|---|---|---|---|
| | Lifetime (L)/ Current (C) | Prevalence | | Summary of Issues which may impact prevalence | | | | | |
| [67] | National specialist clinic, UK | 474 | 78.40% | M = 30.59 (SD = 11.18) | | Any diagnosis | L | 0.58 | Retrospective review of cases referred for assessment of possible Autism. Clinical sample, Compared with non-Autistic group. Participants with Intellectual Disability were excluded. |
| | | | | | | Anxiety | | 0.39 | |
| | | | | | | Depression | | 0.16 | |
| | | | | | 0 | Psychosis | | 0.02 | |
| | | | | | | OCD | | 0.18 | |
| | | | | | | ADHD | | 0.1 | |
| | | | | | | Bipolar Disorder | | 0.08 | |
| [1] | Census Data, Scotland | 6649 | | 25+ | 29.4 | Any diagnosis | L | 0.33 | Large, representative sample compared with people without autism, Self-reported diagnosis |
| [68] | Census Data, Scotland | 7715 | | 16–24 | 18.1 | Any diagnosis | L | 0.15 | Large representative sample compared with people without autism, Self-reported diagnosis |
| [69] | Medicaid Data (2008–12), USA | 166952 | 74.15 | 18–64, M = 28.28 (SD = 11.12) | 23.53 | Anxiety | L | 24.85 | Large representative sample, which is compared with random sample of general population, |
| | | | | | | ADHD | | 20.14 | |
| | | | | | | Bipolar Disorder | | 25.02 | |
| | | | | | | Depression | | 27.45 | |
| | | | | | | OCD | | 7.94 | |
| | | | | | | Other Psychoses | | 12.33 | |
| | | | | | | Schizophrenic Disorders | | 11.23 | |
| [70] | Population based registry data, Norway | 7528 (Autism) | 72.10 | M = 26.2 (SD = 7.9) | | Anxiety | L | 0.14 | Large population-based sample compared with other conditions and remaining population. |
| | | | | | | Depression | | 0.14 | |
| | | | | | NA | Psychosis | | 0.07 | |
| | | | | | | Bipolar Disorder | | 0.03 | |
| | | 1467 (Autism + ADHD) | 71.20 | M = 26.8 (SD = 7.1) | | Anxiety | L | 0.21 | |
| | | | | | NA | Depression | | 0.2 | |
| | | | | | | Psychosis | | 0.07 | |
| | | | | | | Bipolar Disorder | | 0.06 | |

ID = Intellectual Disability, M = Mean, SD = Standard deviation, ADHD = attention deficit hyperactive disorder, CPRD = Clinical Practice Research Datalink, DSM = Diagnostic and Statistical Manual, FXS = Fragile X Syndrome.

reviews. Five reviews were focussed on pharmacological interventions, nine examined employment focussed interventions, seven reviewed the evidence for psychological therapies, and the remaining 12 explored evidence for mixed interventions and approaches. Critical appraisal was conducted using AMSTAR2 [42]. Four reviews were rated high, four were moderate, five were rated low and 18 critically low, indicating strong risk of bias (see Table 3).

Exceptionally, one paper described the inclusion of autistic researchers within the research process [71]. This study included a community council comprising 18 people who mostly identified as autistic or were the parent of an autistic adult, and were researchers, medical or

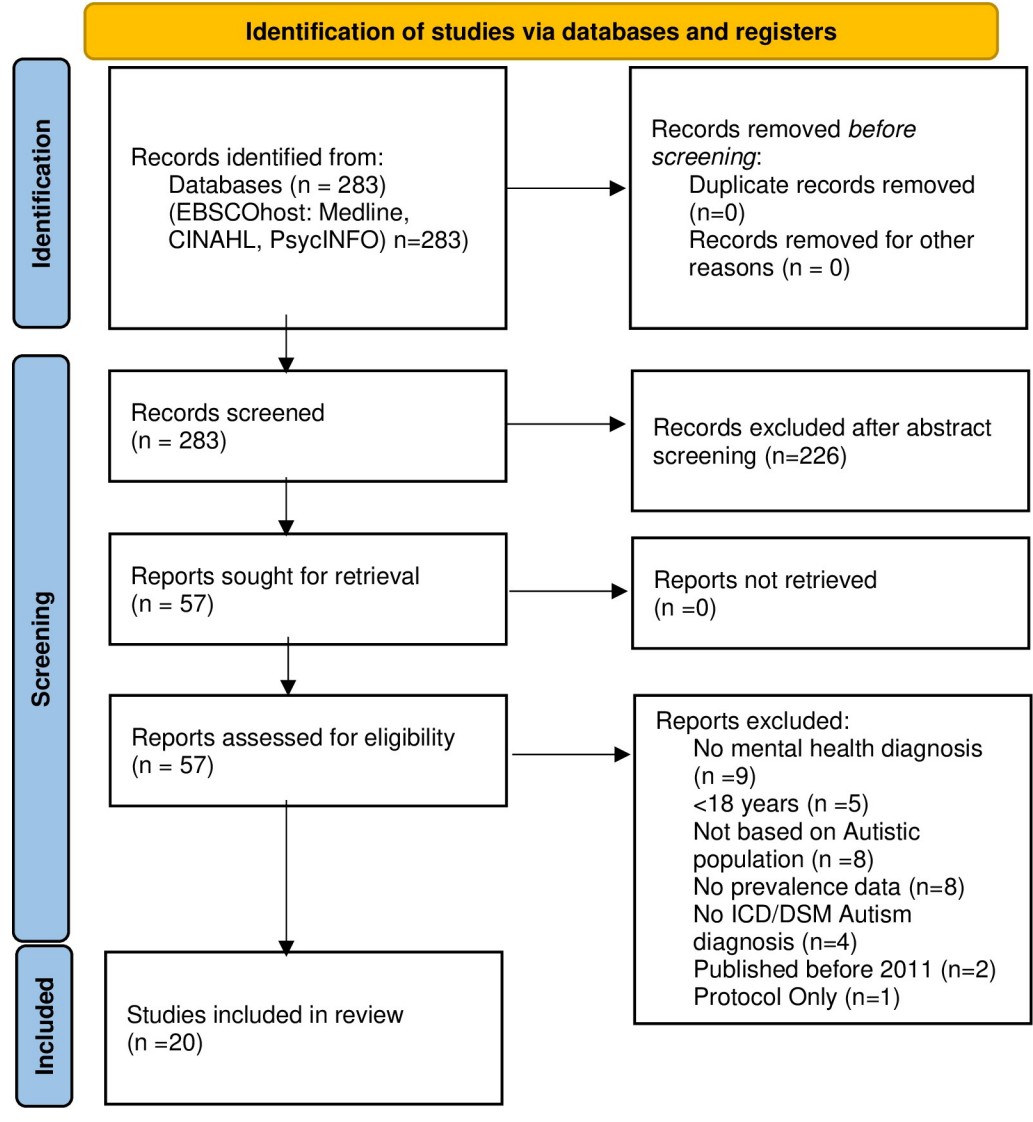

**Fig 1. PRISMA flowchart for prevalence studies.**

mental health professionals, authors, or advocates. This council reviewed study results and contributed to study recommendations [71].

Heterogeneity in outcome measures and variation in the content, length, and delivery of interventions prevented pooling. Outcome measures were often not referenced adequately to permit investigation into their reliability or validity for this population. Full name and authors of outcome measures where reported are included in S3 File. Reviews did not always report results of individual outcome measures and often used diagnostic assessment tools as outcome measures which are not only insensitive to change but indicate a focus on the reduction of core autistic features [72]. Few interventions were manualised, and there was limited reporting on the training, skills, or experience of practitioners.

**Summary of evidence for effectiveness and acceptability of interventions.** The summary synthesis of intervention sub-categories with GRADE recommendation is presented in Table 4. There was considerable overlap in the primary studies reported within the retrieved

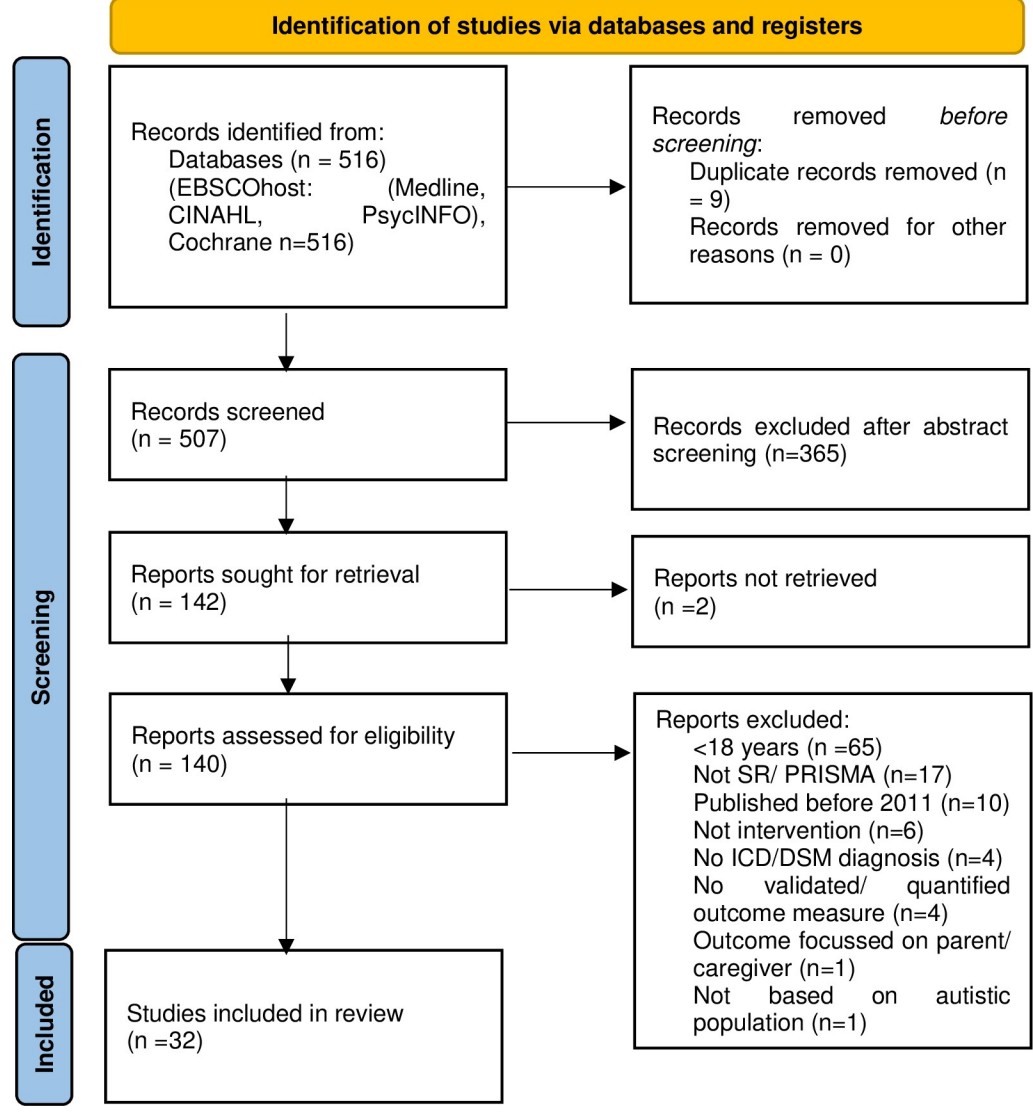

**Fig 2. PRISMA flowchart for intervention studies.**

systematic reviews (S4 File) which prevented further data synthesis. Column 3 of Table 4 outlines factors which may impact the acceptability of interventions to autistic adults including research limitations, indications of adverse effects, adverse outcomes, or priorities contradicting those identified by the autism community. Overall, results indicate a need for further robust research. None of the included interventions were rated as 'evidence based', and eight were 'Not Recommended.' There was 'Unestablished' or 'Emerging Evidence' for the remaining interventions.

**Pharmacological interventions.** Five reviews [81, 88, 89, 98, 100] considered 139 studies evaluating pharmacological intervention for autistic individuals. One review was high quality (Table 3). Managing behaviours with medication as a first line of intervention or using medication including SSRIs (Selective Serotonin Reuptake Inhibitors) or Oxytocin for core features of Autism is not recommended (Table 4) [46, 100]. However, there was emerging evidence for use of medication as a last line of intervention. Oxytocin may offer some benefit but did not

**Table 3. Systematic reviews of interventions for autistic adults.**

| Study | Intervention | Included Studies | Population Characteristics | Ability Level | Outcome Measures | Condition Targeted | Effectiveness, Reported effect sizes for Data Synthesis | Negative Consequences | AMSTAR2 Rating [39] |
|---|---|---|---|---|---|---|---|---|---|
| [71] | Interventions to address health outcomes | 19 studies; 4 RCT (2 CBT, I Mindfulness, I PEERS) | Age: 17–44 (M = 22.3, SD = 7.65) Gender: Male = 99, F = 45: 42% of studies included only male samples | 5 studies– 100% ID, 7 studies 0% ID, 3-not reported, 4 = unclear | NA | Physical and mental health | Mixed; NE; Emerging evidence for cognitive behavioural approaches for improving self-reported mood and anxiety symptoms, Emerging evidence for mindfulness to address self-reported health outcomes of depression and anxiety amongst autistic adults without ID, unestablished evidence for medical interventions and ECT. | Negative response to ECT with worsening symptoms, some autistic people report CBT is unhelpful for them, Mindfulness not likely implemented in the same manner as researched in this review, some autistic people see social skills interventions as teaching camouflaging- Social skills interventions present specific behaviours as negative or wrong, and therefore promote feelings of shame as related to autistic features that are part of identity. Side effects of medication and dosing are not well evaluated. Vocational intervention research does not consider health/ quality of life. | H |
| [73] | Psychosocial Interventions including ABA, Social Cognition Training, PEERS programme, Community Based Interventions | 13 studies; 4 RCT (3 social cognition training; 1 other) | N = 1–65 Mean Age range: 18–36.27 6 studies males only (88.2% Male) | NA | NA | Autism | +, NE; social cognition training was associated with improvement in face and voice recognition, theory of mind skills, and social communication skills. | Study outcomes included repetitive behaviour and deficits of social interaction | CL |
| [74] | Acceptance Commitment Therapy (ACT) | 8 studies; 1 RCT | N = 54, Age: M = 16.6- M = 49, (M = 16.6 years for RCT population) | 4 studies- 100% ID, 4 studies–unclear. | AAQ-9, CFQ-7 | Psychological health | Mixed, NE. | NA | CL |
| [75] | Cognitive Remediation Interventions | 13 studies; 4 RCT(3 included adults), 4 case series, 2 feasibility studies | N = 1–109; Age: M = 18-M-49.5. Gender: 202 (73.7%) Male | NA | International Affective Picture System, Corsi-BTT, BRIEF, SSRT, N-Back task, Gender-emotion switch task, BACS-J, WCST, CPT, ScoRS-J, LASMI, MCCB, WCST, MSCEIT, PERT, PEDT, PEAT, SCS | Facial Affect, working memory flexibility, core-cognitive, employment outcomes | Mixed, NE | NA | CL |
| [76] | Psychoeducational Interventions including recreational therapy, behavioural techniques, multisensory room. | 56 studies; 4 RCT (1) Behavioural Intervention; (2) Leisure program; (3) Vibroacoustic chair and music therapy; (4) Residential program based on TEACCH | Adults | Level 3 Autism | NA | Behavioural outcomes including self-injurious behaviour, emotional functioning, aggressive/destructive behaviours | Mixed; NE | NA | Mod |
| [77] | Group based social skills training | 18 studies; 5 RCT; (1) social skills; (2) PEERS; (3) JOBSS; (4) PEERS; (5) PEERS | Adults with ASD following DSM-V criteria, mean age>18 years, 68.4–98.1% Male, | Overall IQ>90 (90–100). | SRS, RMIE. WFIRS-S, BRIEF-A, ECSI. VABS-S, SFQ, SES, GAD, AQ, EQ, DAQ, SRSS, TYASSK, SELSA, ToM, RMET, IPR, STAI-A, BDI, IPR, RSES, UCLALS, CCAPS, QSQ, MSCS-C, DASS-21, SSIS-RS, LSAS, Patient health Depression Scale, Vineland Social Functioning, ABA-Adult, PESE, PSSE, SSPA, SCSQ, | Autism | + effect for social skills training over control 0.93[95% CI 0.55–1.30, Q-test, $p = 0.39$] | NA | CL |
| [78] | Interventions for improving employment outcomes including Project SEARCH, Virtual reality | 3 studies; 3 RCT(1 Project SEARCH, 1 Virtual Reality Job Interview Training, 1 Modified Project SEARCH) | N = 108 Age: M = 19.13- M = 25 | NA | Employment status | Autism | +, NE | NA | Mod |

(*Continued*)

**Table 3.** (Continued)

| Study | Intervention | Included Studies | Population Characteristics | Ability Level | Outcome Measures | Condition Targeted | Effectiveness, Reported effect sizes for Data Synthesis | Negative Consequences | AMSTAR2 Rating [39] |
|---|---|---|---|---|---|---|---|---|---|
| [79] | Mindfulness | 10 studies; 2 RCT(1 Mindfulness with caregivers, 1 Mindfulness with ASD adults), 8 quasi experimental. | N = 454 (74 children, 139 adults) Adult's age: M = 38.4 (SD = 10.3) Gender: Male = 217/ F = 237 | 5 studies excluded IQ<85 | DERS, OQ, AQ, MAAS-A, PSWQ, RRS, WHO-5, SRS, FFMQ, IM-P, PS, PSI, PSI-SF, GHQ, FMI, PSS, FQOL, CBCL, SCL90R, RRQ, GMS, WHO QOL CERQ, DASS-21, MAAS, CAMM, ASEBA, WHORRS, CSQ-CA, CSRQ, SCS, STAI, POMS, ESS, BDI, ZBI, ASQ, HADS, GMS, SRS-A, ISI, MAAS, | Psychological distress, wellbeing in Autism | +, SMD (k = 1) -adults post-intervention g = 0.87[95% CI 0.65, 1.09] | NA | H |
| [80] | Employment programmes and interventions | 10 reviews and 50 empirical articles | N = 58134 Age: 15–65 Gender: Male = 74.91% | IQ = 30-164 (M>70), 43 studies = NA | NA | Autism | Mixed; NE | Lack of support and understanding in the workplace, overrepresentation in low-paid, casual, and entry-level positions. High costs to families in terms of time, loss of income, loss of career opportunities and depreciation of work skills while supporting family members. | CL |
| [81] | Treatment of aggression | 70 studies; 21 case reports, 17 NRCT; 16 prospective open trials; 8 retrospective reviews, 1 naturalistic case-control study, 7 RCT; (1) Vigorous antecedent aerobic exercise, (2) Fluvoxamine 50-300mg/day for 12 weeks, (3) Risperidone up to 6mg/day for 12 weeks, (4) Risperidone low (2mg/day) or high (4-5mg/day for 4 weeks, (5) Vibroacoustic Music, (6) Transdermal Nicotine, (7) dextromethorphan/quinidine. | N = 1–61, Age: Adult subjects, Gender: Mixed Male/F | NA | Visual analogue scale, number of reported incidents, ABC, CGI-I/S, PANSS, BAS, SIB-Q, ABC-I, Behaviour Problems Inventory. | Aggression | Mixed; NE | Weight gain, constipation, metabolic syndrome, tachycardia, activation of target symptoms, weight loss, decrease in cholesterol, decrease in triglycerides, hair loss, sedation, buccal numbness, elevated liver enzymes, difficulty waking in the morning, increased appetite, daytime drowsiness, rhinitis, gynecomastia, anxiety, agitation, akathisia, pedal oedema, nausea, nightmares, seizures, skin picking, low Heart Rate, low Blood Pressure, increase in aggressive behaviour, | CL |
| [82] | Transition and vocational Interventions | 35 articles; 39 studies and 8 case studies, No RCT | N = 1–100; Age: 13–55 years, Gender: Mixed Male/F, | 11 studies = no ID, 13 studies = ASD +ID, 9 studies = Mixed ASD+ID and ASD only, 6 studies = NA | NA Employment/ Task related performance | Autism | Mixed | NA | CL |
| [83] | Support for adults with Autism; social skills training, job interview training, music, and dance | 32 studies; 8 NRCT; 15 uncontrolled (one-group); 9 RCT (1) PEERS, (2) Group CBT, (3) PEERS, (4) PEERS, (5) Group social-cognitive programme, (6) Psychoeducation, (7) social skills, (8) Virtual reality job interview training, (9) Multimedia Interview training | N = 3–100 (Median N = 13.5); Age: M = 25, Gender: 80% Male | No ID | Self-report/ Observation, Index of Peer relations, SSRS, VABS, OAWP-SR, SRS, ASQ, DAQ, EQ, SPS, CSSCEI, SELSA, QSQ, HADS, QoLI, SCS, RSES, SCL-90, BDI, Adult ADHD-SR, CGI, IPR, STAI, SSPA, SSIS, LSAS, SPI, SPSI, SCSQ, SSPA, CASS, BDEFS, SACQ, PHQ, ISRI, HSI, QMT, EES, MET, IRI. | Autism | Mixed, NE, Results suggest job interview training may be effective in improving interview performance, social skills training can be effective in improving self-rated social skills, autism symptoms and social relations; and employment programmes can be effective in increasing employment for autistic adults without ID. The evidence for other interventions and outcomes is inconclusive and limited. | It is unclear how outcomes such as improved mock interview performance might generalise to real world outcomes. | L |

(*Continued*)

**Table 3.** (Continued)

| Study | Intervention | Included Studies | Population Characteristics | Ability Level | Outcome Measures | Condition Targeted | Effectiveness, Reported effect sizes for Data Synthesis | Negative Consequences | AMSTAR2 Rating [39] |
|---|---|---|---|---|---|---|---|---|---|
| [72] | Music therapy | 36 studies, only 3 with adult participants, No RCT | N = 1251, 22 autistic participants aged 13–29 years 30 autistic participants aged 9–21 years 7 autistic (from 9 participants) aged 13–20 years | NA | ADOS, FEAS, ADI-R, SSRS, PDDBI, ESCS, IPR, STAI, SRS, ATEC, ECA-R, AQR, TEA-Ch, CARS2-HF, 9 behavioural tasks, CCC-2, SRS-II, PPVT-4, refMRI, K-WISC-IV, Korean Social Skills Rating System, Soundscape Questionnaire, self-report | Autism | Mixed, NE, most studies failed to demonstrate effectiveness | NA | CL |
| [84] | Treatment of depression including CBT, vocational skills, social skills, remediation therapy, mindfulness | 25 articles; 7 RCT- 3 with adult participants: (1) IPT +CBT; (2) Group Vocational Skills Training; (3) Group MBT, 3 non-randomised studies, 4 case study, 9 open-label, 2 case series | N = 1–52, Age: 6–65 years; Gender: M/F | IQ = 82-118 (M = 99) | CDI-2, BDI-II, CCAPS-34, SCL-90-R, RCADS, BASC-2, CDI, DASS, PHQ-9, HADS, HAM-D, CDRS, MADRS | Depression | Mixed, NE, CRT did not improve symptoms of depression in autistic adults; Behavioural Therapy study did not include adults; CBT results were inconsistent, and the review was unable to make clinically useful conclusions; Combined psychosocial intervention did not find a positive treatment effect; Mindfulness showed a positive treatment effect for depression in autistic adults without ID; Inconsistent results for social/academic/vocational skills training for autistic adults; Phenytoin has positive treatment effect for depression in autistic adult (n = 1); NMDA receptor antagonist showed positive effect treating depression in autistic adult (n = 1), | Irritability, insomnia, decrease in appetite, abdominal pain, headache, dizziness, sedation, extrapyramidal side effects, numbness of limbs and face, blurred vision. | CL |
| [20] | Social Skills Interventions including CBT, PEERS, SUCCESS, ASSET, Joint attention | 26 studies; 1 qualitative, 6 mixed method, 7 single-subject, 6 quasi-experimental, 6 RCT: 3 PEERS; 1 Job-based social skills group; 1 Interview Skills group; 1 ACCESS Program. | N = 342 (2–49); Age: 16–55 years; Gender: 77.6% Male | 15 studies IQ means ranged from 93.38–113.30, 1 study IQ>80, 1 study = average/ above average IQ, 9 studies = NA, 5 studies = college students | GPA, BRIEF-A, D-KEFS SSPA, SRS-2, SFQ, SRS, GSE, PESE, PSSE, PHQ-9, GAD-7, SSRS, SELSA, EQ, QSQ, SSI, TYASSK, RMET, ERQ, BPAQ, SPAI-23, BDI-II, IPR, STAI, AQ, EQ, SES, UCLA Loneliness Scale, CCAPS-34, Social Validity Questionnaire, ASES, Autism Awareness Scale, DIOS, SRS-A, TONI-3, ACS-SP, Ekman 60, Triangles, TYASSK, ER40, TA-SIT, The Hinting Task, HCAS, VABS-II, SSIS-RS, SRS, QSQ-YA, LSAS-SR, SPIN, VBAS, SPS, ABAS-3, Seven Component SD Skills Survey, CSES, ASEBA-ASR, URP-ASSET, FEIT, SQSQ, ACS-SP, Social Attribution Task | Autism | Mixed, NE, 4 RCTs showed positive changes to caregiver reported outcome measures, individual studies showed no effect on loneliness, social anxiety depressive symptoms or coping self-efficacy. | NA | CL |
| [85] | Video based interventions for employment skills | 19 studies; 14 single subject design, 5 group design. | N = 164; Age: not stated, except studies must include at least one participant over 16; years, Gender: Male = 85.4%/ F = 14.6% | NA | NA | Autism | NE | Reported outcomes included Physical appearance, greeting customers, Communication skills for job interviewing, interacting with co-workers. It is unclear how these relate to employment or quality of life | CL |
| [86] | Vocational support | 10 studies; 3 single-case, 3 interventions studies, 2 longitudinal, 1 programme evaluation, 1 routine data | N = 3–382221; Age: 18+ | NA | NA | Autism | NE | Autism severity increased in participants in sheltered employment, reported outcomes included work behaviour | CL |

(*Continued*)

**Table 3.** (Continued)

| Study | Intervention | Included Studies | Population Characteristics | Ability Level | Outcome Measures | Condition Targeted | Effectiveness, Reported effect sizes for Data Synthesis | Negative Consequences | AMSTAR2 Rating [39] |
|---|---|---|---|---|---|---|---|---|---|
| [87] | Psychosocial interventions for employment including Project SEARCH | 10 RCT: (1) Family centred transition planning; (2) iPod touch based vocational support; (3) Project SEARCH; (4) Multimedia employment training; (5) Group social skills; (6) VR Job interview training; (7) Peer model videos; (8) Robot mediated mock interview; (9) Interview training using robot; (10) CET | N = 423, Age: M = 17.6–24.5 years | 4 studies IQ>70, I study = at least 6th grade reading level, 1 study-6% beyond high school diploma, 24% high school certificate, 2% less than high school, 2 studies = NA | Confidence Rating Scale, Observation, employment status, wage. | autism | +, Employment within 6 months of intervention: communication skills training (k = 2) (RR 2.27, 95% CI [0.73, 7.07] $I^2$ = 0%, p = 0.16); Vocational Support (k = 1) (RR 11.57 95% CI [2.84, 47.20], p = 0.0006); Worked hours within I year of intervention (k = 1) (SMD 14.30 95% CI [11.40, 17.20], p<0.00001). | Outcomes included 'could get a job', rather than actual employment | CL |
| [88] | Oxytocin | 7 RCT | N = 101 autistic participants; Age: M = 11.2-M = 33.2; Gender: Male = 95 | 6 studies IQ>70, I study = Fragile X. | RMET, CGI, Diagnostic Analysis of Nonverbal accuracy, UNSW Facial Emotion Task, | Autism | Mixed, NE, Results of individual studies indicate reduction in repetitive behaviours and self-injury; increased eye gaze, improved nonverbal communication. | Drowsiness, anxiety, depression, headache, tingling, backache, trembling, restlessness, stomach cramps, enuresis, sweating, irritability, allergy symptoms, fatigue, leg shaking, increased energy | CL |
| [89] | Opioid Antagonists | 10 RCT: (1) Naltrexone 0.5, 1.0, 1.5, 2mg/kg; (2) naltrexone 50mg, 100mg; (3) Naltrexone 0.5, 1.0, 2.0 mg/kg; (4) Naltrexone 50mg; (5) Naltrexone 0, 25mg, 50mg, 100mg; (6) 0.5, 1.0, 2.0mg/kg; (7) Naltrexone 1.5mg/ kg; (8) Naltrexone 50mg, 100mg; (9) Naltrexone 100mg single dose, cohort 1 50mg, cohort 2 150mg; (10) Naltrexone 50mg. | N = 124 (n = 49 with autism), Age: 14–67 years, Gender: Male = 91/ F = 33 | All participants had ID | PTQ, BDC, FAIR, Paired Associates Test, SOME, VABS, CGI | Autism, ID | Mixed, NE, 8/10 studies found reduction in self-injury, statistically significant in 6. | Weight loss, mild liver function test abnormalities, loss of appetite, thirst, yawning, nausea, tiredness, sedation | CL |
| [90] | Competitive integrated employment | 25 studies; 6 RCT; 1 VR Interviewing; 1 Personal Digital Assistant; 4 Project SEARCH with ASD supports; 4 quasi-experimental; 13 secondary data analysis | N = 5–49623 | NA | Employment Status including volunteer positions | Autism | Mixed, NE, 73.4%-90% of participants undertaking Project Search + ASD supports achieved competitive integrated employment. Sheltered workshops did not support competitive employment | Project SEARCH +ASD Supports used ABA instructional strategies | CL |
| [91] | AIT | 7 studies; 5 parallel design trials, 2 cross-over studies, No RCT | N = 10–80; Age: 3–39 | NA | Not specified but included tests of cognitive ability, core features of autism, hyperacusis, auditory processing, behavioural problems, attention and concentration, activity level, quality of life in school and home environments, adverse events. | Autism | Mixed, NE | Minor physical complaints, minor side effects, potential harms of AIT include whether machine output levels exceed safe limits and risk hearing loss. | H |
| [92] | Group Social Skills | 5 studies; 2 quasi-experimental comparative trials, 3 single arm interventions, No RCT | N = 10–49; Age: M = 25.8, 1–55 years; Gender: Male = 85% | All available participant IQs were in the average range. | SELSA, QSQ, SSI, EQ, TYASSK, SRS, SSRS, AQ, IPR, BDI, STAI, FEIT, SSCQ, SSPA, | Autism | Mixed, NE, no study discussed clinical significance of change in outcome measures although study findings were generally positive. | NA | L |

*(Continued)*

**Table 3.** (Continued)

| Study | Intervention | Included Studies | Population Characteristics | Ability Level | Outcome Measures | Condition Targeted | Effectiveness, Reported effect sizes for Data Synthesis | Negative Consequences | AMSTAR2 Rating [39] |
|---|---|---|---|---|---|---|---|---|---|
| [93] | CBT | 6 studies; 2 RCT: (1) CBT for OCD, (2) Mindfulness based group, 1 quasi-experimental, 1 case series, 2 case study | N = 105 (1–41), Age: 16–65 | NA | SPAI, LSAS, BDI-II, YBOCS, BDI, BAI, CGI, WSAS, SCL-90-R, RRQ, DGMS, | Autism, Depression, anxiety, low mood, alcohol, OCD, agoraphobia, PTSD | +, NE, 1 RCT showed no significant effect for CBT for OCD; 1 RCT showed improvements in anxiety ($d = 0.76$) and rumination ($d = 1.25$) following mindfulness group. | CBT reported to be unhelpful. | CL |
| [94] | CBT for social anxiety | 4 Single case studies, No RCT | N = 4; Gender: Male = 4, Adult = 3. | NA | SCID IV, LSAS, STAI, BDI II, CGI, RCMAS, FSSC, VABS II, BASC 2PRS, SPWSS, RSE, CORE OM, BSI, | Autism, Social Anxiety and Depression | +, NE, general reductions in anxiety. | NA | CL |
| [95] | Family Therapy | 0 studies | NA | NA | NA | Autism | NA | NA | NA |
| [96] | Non-pharmacological; social functioning and language skills, vocational rehabilitation; cognitive skills training, independent living skills | 41 RCT (20 Social Functioning & language Skills Interventions; 10 Vocational Rehabilitation; 11 Cognitive Skills Training; 1 Independent Living Skills) | N = 846 autistic adults in intervention group, 819 in control groups; Gender: Male = 610/ F = 270 | High heterogeneity of intelligence measures used and participants IQ scores. | SPQ, RMET, SRS-2, WFIRS-S, ASR, UCLAS, MCCB, MSCEIT, PERT, SCP, SRS, MRAI, BTFR, WMS, ERP, GAD, LSAS, STAI-T, HADS-D, CORE-OMU, SSRS, SELSA, EQ, QSQ, SSIS, TYASSK, QOL, SSS, ASD-DA, VABS, MESSIER, PWI-ID, DEX, CANTAB, CHART, SIS, EPER, QOLI, SOC, RSES, SCL-90, AQ, BDI, ASRS, CGI-S, CGI-I, DASS-21, CFQ, BAFT, CEEQ, IRI, ADOS-G, QSQ-YA, LSAS-SR, SRS, BACS-J, WCST, CPT, GAF, WCST, ScoRS-J, LASMI, GSE, SPS, PHQ-9, SBS-5, ADI, IGIRT, TASIT, MIRI, GRADE, SCQ, VABS-II, SDS, QoL-Q- Abridged, ABAS3, SDSS, CSES, ASR, MASC, MRI, YBOCS, DYBOCS, OCI-R, BAI, WSAS, SCAS, CHOCHI-R, FAS-PR, SCL-90-R, RRQ, GMS, SRS, MASC, ERSES, DASS, ERS, AAPEP, ABI, SIS, BDEFS, SACQ, CGI-I, ABCL, PSS, SRS, | Autism | Mixed, NE, Individual studies produced positive results, but heterogeneity of outcomes prevented data synthesis | Outcome measures included "ASD Symptoms", 'comprehension of irony. Vocational study outcomes included 'interview skills' which doesn't reflect employment or QoL. | Mod |
| [97] | Vocational Interventions | 5 studies; 1 NRCT, 2 prospective cohort studies, 1 case series, 1 cross-sectional study, No RCT | N = 1999 | 1 study = NA, 4 studies IQ = 41.14–110.7 | NA | Autism | Mixed; NE | Participants in sheltered workshop intervention experienced increased autism symptom severity. | CL |

(*Continued*)

**Table 3.** (Continued)

| Study | Intervention | Included Studies | Population Characteristics | Ability Level | Outcome Measures | Condition Targeted | Effectiveness, Reported effect sizes for Data Synthesis | Negative Consequences | AMSTAR2 Rating [39] |
|-------|-------------|------------------|---------------------------|---------------|------------------|--------------------|---------------------------------------------------------|----------------------|---------------------|
| [98] | Psychopharmacologic Interventions | 43 studies; 4 RCT | N = 347; Age: M = 29.12 (SD = 5.72); Gender: Male = 174/ F = 35 | 142 (40.92%) participants had ID. | CGI, Y-BOCS, SIB-Q, Ritvo-Freeman Real-Life Rating Scale, Vineland Maladaptive Behaviour Subscale, Y-BOCS-Compulsion, ABC- irritability, Hamilton Anxiety Scale, Brown Aggression Scale, CARS, ESRS, DOTES, GAF, SIB-Q, Connors Rating Scale, DISCUS, NSEC, | Behavioural disturbance in autistic adults | Mixed, NE, No established evidence base to support SSRIs to reduce repetitive behaviours; No evidence to support venlafaxine to reduce repetitive behaviours or ASD severity; No evidence based for atomoxetine to reduce hyperactivity and impulsivity in ASD; limited evidence to support use of clomipramine in autistic adults; antipsychotic agents are not efficacious for reducing challenging and repetitive behaviour in autism, there is unestablished evidence for olanzapine to reduce behavioural disturbance in autistic adults; there is promising evidence for risperidone to treat behavioural disturbance in autistic; there is a need for further research into the effects of aripiprazole on behavioural disturbance in autistic adults, evidence base for other antipsychotic medications is limited, Limited evidence to support use of divalproex sodium/ sodium valproate to reduce behavioural disturbances in autistic adults; limited evidence for propranolol in treating challenging behaviours in autistic adults; Lorazepam had no effect on aggressive behaviour; Buspirone reduced self-injurious behaviour in a single case study; Limited evidence for Naltrexone to reduce self-injurious behaviour in autistic adults; Clonidine is associated with reduced aggressive behaviour in 2 studies; Methylphenidate may be indicated for autistic adults with co-occurring ADHD. | Agitation, self-picking, syncopal episode, anorexia, headache, tinnitus, alopecia, weight gain, sedation, bad or vivid dreams, insomnia, dry mouth, hyperactivity, racing thoughts, irritability, nausea, sedation, drowsiness, abdominal cramping, seizures, behavioural difficulties, abnormal gait, sialorrhea, enuresis, dyspepsia, diarrhoea, constipation, gastrointestinal complaints, oculogyric crisis, akathisia, restlessness, weight loss, gynecomastia, rapid heartbeat, shaking, vomiting, nosebleeds, catatonia, tachycardia, lethargy, dystonia, depression, elevated liver enzymes, difficulty waking in the morning, hair loss, buccal numbness, fatigue, increase in self-injurious behaviour, side effects of clomipramine may outweigh any treatment gains, risperidone is associated with notable side-effects particularly weight gain. | L |
| [99] | CBT | 48 studies; 24 studies focussed on ASD symptoms or features; 24 studies examined effectiveness of CBT for affective disorders–only 4 of these included adult participants. | Age Range: 5–65 years | NA | BAI, Y-BOCS, SCL-90-R, RSES, LSAS, HAM-A, OCI-R, CHOCI, DASS, Strengths and Difficulties Questionnaire, SSPA, SCSQ, SELSA, SRS | Autism, Affective disorder | +, Random effects meta-analysis estimate treatment effect of CBT for symptoms of affective disorders in autism based on self-report measures $g = 0.24$ [95%CI -0.05, 0.53], $z = 1.6$, $p = 0.11$, $I^2 = 69\%$. Informant reported outcome measures $g = 0.66$ [95% CI 0.29, 1.03], $z = 3.49$, $p<0.001$, $I^2 = 78\%$; Clinician Rated Outcome Measures $g = 0.73$ [95% CI 0.38, 1.08], $z = 4.05$, $p<0.001$, $I^2 = 69\%$. | 24 studies focussed on symptoms or features of autism including social skills, theory of mind, facial emotions, and affectionate communication; Group CBT may not be associated with greater effectiveness as therapists may be unable to tailor interventions to individual needs. | L |
| [100] | SSRI | 9 RCT: 4 included adults (1) Fluoxetine; (2) Fluoxetine; (3) Fluvoxamine; (4) Citalopram. | Age: 18–60 years | For the 4 adults studies: 1 study-IQ = 53 to 119, 1 study IQ>70, 1 study 92% IQ>70, 1 study included intellectually able and disabled adults. | CGI/I/AD, CY-BOCS, CY-BOCS-PDD, Y-BOCS | Autism | +, Proportion Improved for CGI-I RR = 12.58 [95% CI 1.77, 89.33] $z = 2.53$, $p = 0.01$ $I^2 = 0$; Evidence from small studies with unclear risk of bias indicates significant improvements in clinical global impression (fluvoxamine, fluoxetine), obsessive-compulsive behaviours (fluvoxamine), anxiety (fluoxetine) and aggression (fluvoxamine). | Apathy, sedation, decreased sexual interest, flatulence, nausea, sedation, upper gastrointestinal disturbance. | H |
| [101] | Interventions targeting expressive communication | 22 studies; 8 RCT (3 PEERS, 2 executive function program, 1 ACCESS, 1 VR Job training, 1 Oxytocin); 14 Single case design. | N = 256; Age: 18–43; Gender: Male = 192(75%) | 6 of RCT studies IQ>70, 1 study = participants had limited spoken language, 1 study = NA, 2 SCD studies–participants minimally verbal, Other SCD studies reported IQs ranged from 42 to 72. | SRS, SSRS, SSIS, MESSIER, LASMI, ABAS. BACS-J, ADOS | Expressive communication | +, PEERS SRS scores (k = 3) SMD = 0.825, SE = 0.221, $var = 0.049$ [0.392, 1.259] $z = 3.732$, $p = 0.000$; PEERS SSRS scores (k = 3) SMD = 0.473, SE = 0.217, $var = 0.047$ [0.048, 0.898] $z = 2.183$, $p = 0.029$ | Interventions included ABA, | L |

RCT = randomised controlled trial, M = Mean, SD = standard deviation, F = female,+ = positive effect, - = negative effect, H = high, CL = critically low, L = low, Mod = moderate, NA = not available, CBT = cognitive behavioural therapy, SSRI- selective serotonin reuptake inhibitor, NRCT = non-randomised controlled trial, ECT = Electro-convulsive Therapy, AIT = Auditory Integration Training, ACT = Acceptance and Commitment Therapy, ABA = Applied Behaviour Analysis, ASSET = Assistive soft skills and employment training, ID = intellectual disability, IQ = Intelligence Quotient, SCD = Single Case Design, ASD = Autism Spectrum Disorder, CET = Cognitive Enhancement Therapy, EST = Enriched Supportive Therapy, NE = no effect size calculated, OR = Odds Ratio, SE = Standard Error, SMD = Standardised Mean Difference, MA = Meta-analysis, OCD = Obsessive Compulsive Disorder, VR = Virtual Reality, PEERS = Program for the Education and Enrichment of Relational Skills.

affect global clinical status [88]. Risperidone may be useful in the management of repetitive, aggressive, and self-injurious behaviour [81], although side-effects are problematic [98]. There was limited evidence to support the use of opioid antagonists to reduce self-injury in autistic adults [89]. However, fluoxetine or fluvoxamine may be useful in the management of repetitive and obsessive-compulsive behaviour and anxiety where other interventions are not available or possible due to the individual's level of distress or aggression [98]. Overall, there is a need for future research to consider the acceptability of pharmacological interventions including further investigation of side-effects.

**Employment focused interventions.** Nine reviews of evidence for employment focussed interventions considered 100 unique publications [78, 80, 82, 83, 85–87, 90, 97]. None of the reviews were high quality (Table 3). Reviews revealed emerging evidence that supported employment including Individual Placement Support (IPS) and Project Search, yields positive outcomes for autistic people [78, 80, 82, 86, 87, 90, 97]. Notably, autistic adults, undertaking Project SEARCH with autism support were eleven times more likely to achieve employment than those attending special education [87]. However methodological concerns mean this result must be interpreted with caution as studies did not include comparable control groups or consider participant attrition [78]. Evidence for technology-supported interventions such as virtual reality training was unestablished as the relationship to paid employment was not confirmed [85]. Employment related social skills training research often focussed on alternative outcomes to employment status, such as interview skills performance, and therefore the evidence for such an approach is unestablished. Sheltered workshops were not recommended as they were not associated with supporting autistic people into employment but could provide other benefits. Further research is required to consider the impact of employment focussed interventions not only on employment status and wage, but also on quality of life [24].

**Psychological therapies.** There were 7 reviews of psychological therapies including 215 studies [74, 75, 79, 93–95, 99]. Only one review was of high quality (Table 3). The reviews revealed emerging evidence (Table 4) for the use of mindfulness for the reduction of self-reported depression symptoms in autistic adults without intellectual disability [71, 79, 84]. Studies provided emerging evidence for use of Cognitive remediation therapy to improve cognitive function, but small sample sizes and limited follow-up made it difficult to determine meaningful impact or maintenance of any benefit in the longer term [75].

There was unestablished evidence for the use of cognitive behavioural therapy (CBT), although small positive clinical effects on self-reported outcomes were observed [71, 99]. Within nine systematic reviews, which included CBT studies, 11 different types of CBT were described and included CBT combined with other interventions including behavioural techniques, mindfulness, and psychoeducation [93, 94]. These major variations in the intervention provided meant it was not possible to conclude this intervention was effective. Additionally, there were expressed concerns regarding CBT which are outlined in Table 4 and which should be considered in future research.

There was unestablished evidence for family therapy due to limited quality research [95] although non-randomised intervention studies suggest there may be improved knowledge and understanding of core disorder (ASD), and coping styles post-intervention [95]. Acceptance and Commitment Therapy was not recommended due to limited research and insufficient rigour [74] to suggest ACT is effective in the management of psychological distress for individuals with ID [74].

**Mixed interventions and approaches.** Twelve systematic reviews considered 300 studies within 11 sub-categories of intervention identified [20, 71–73, 76, 77, 83, 84, 91, 92, 96, 101]. Two reviews were rated as high quality (Table 3). Evidence for most of interventions in this grouping was unestablished or not recommended (Table 4). However, there was

Table 4. Evidence for interventions.

| Intervention | GRADE level | Acceptability |
|---|---|---|
| **Pharmacological** | | There was no consideration of the acceptability of these interventions to autistic people. One review commented that the autistic community report that medication side-effects and dosing are not well evaluated, and that medication generally, is poorly tolerated [71]. |
| Managing behaviours deemed problematic, with medication, as a last line of intervention | Emerging evidence | Medication may be used to address co-occurring conditions, such as anxiety, self-injury, OCD, or depression [46]. |
| Medication for core features of Autism | Not recommended | Contradicts current guidelines [46]. |
| Managing behaviours deemed problematic, with medication, as a first line of intervention | Not recommended | Contradicts current guidelines [46]. |
| **Employment focused** | | Generally, studies of vocational or employment interventions did not consider health outcomes, or impact on quality of life for autistic adults [71]. |
| Project SEARCH | Emerging evidence | Project Search and other supported employment models are promising models for helping people attain employment, but further research is required. |
| Individual Placement and Support (IPS) | Emerging evidence | Further robust research is required. |
| Supported Employment | Emerging evidence | Cost benefit of supported employment is not known. Some studies reported employee job satisfaction [86]. |
| Technology supported interventions | Unestablished evidence | Technology supported interventions are under-researched and have focussed on proximal measures to employment status e.g., improvement in interview performance, rather than employment outcomes. |
| Employment related social skills training | Unestablished evidence | There is a need to demonstrate the link between these outcomes and employment. |
| Sheltered workshops | Not recommended | No evidence that sheltered workshops support people into employment. They may support other outcomes. |
| **Psychological Therapies** | | Psychological therapies can be used, but with caution, as there is insufficient evidence of their effectiveness over other approaches. There is a lack of evidence over which autism adaptations should be incorporated and how this should be done [90]. |
| Mindfulness | Emerging evidence | Need for more robust research evidence. |
| Cognitive Remediation Therapy | Emerging evidence | Need for more robust research evidence |
| CBT | Unestablished evidence | CBT is not significantly better than alternative interventions and autism adaptations are insufficiently researched [90, 102]. Self-report measures are not reliably associated with significant change following intervention because autistic people may have developmental differences in communication and perspective taking which leads to difficulty reliably reporting symptoms [12, 93, 94]. Reliability and validity of most administered questionnaires has not been established for autism community [93]. Clinical diagnoses in most studies were established using standrad diagnostic criteria despite overshadowing of some mental health diagnoses and autism [93]. There was a wide range of experience with each intervention. One review stated some autistic people report that CBT is unhelpful for them [93]. There is a need for evaluation of long-term impact of CBT interventions [71]. Studies reported effectiveness of CBT for symptoms associated with autism in contravention of current guidelines [99]. |
| Family Therapy | Unestablished evidence | Limited research [95]. |
| Acceptance and Commitment Therapy (ACT) | Not recommended | Limited research and insufficient rigour [74]. |
| **Mixed interventions/approaches** | | Caution is required to ensure interventions do not aim to "cure" autistic traits [44]. |
| Program for the Education and Enrichment of Relational Skills (PEERS) | Emerging evidence | The PEERS approach is acceptable to autistic people where care is taken to avoid shaming non-neurotypical communication styles [68]. |
| Social cognitive interventions | Emerging evidence | There is a need to engage with the autistic community about the concept of theory of mind for describing differences in noticing and interpreting intention, thoughts, or beliefs of others. Additionally, there is a need for clarity about when, why, or how such interventions might be relevant. |

(*Continued*)

**Table 4.** (Continued)

| Intervention | GRADE level | Acceptability |
|---|---|---|
| Social skills interventions | Unestablished evidence | There is a need to engage with the autistic community in relation to strengths-based approaches within social skills interventions to ensure that they are not trying to 'fix' or 'cure' individuals [20]. Social skills are important for relationships however, some autistic people see some social skills interventions as teaching camouflaging, which has been associated with suicidality [71, 103]. Some social skills interventions present specific behaviours relating to autistic features as wrong which can be detrimental to the health or identity of autistic people [71]. Reported study feedback includes statements that interventions were helpful [20]. Another study reported participant feedback that social skills groups were acceptable, and they were able to put some of the skills into practice. Reported attrition rates ranged from 10–18% [92]. |
| Communication Interventions | Unestablished evidence | Limited research evidence |
| Music and dance therapies | Unestablished evidence | Some studies targeted core symptoms of autism [72]. Limited evidence of effectiveness in autistic adults. |
| Environment, leisure & participation interventions | Unestablished evidence | Limited research and insufficient rigour |
| Behaviourist approaches | Unestablished evidence | There have been concerns raised regarding Applied Behaviour Analysis (ABA) by autistic adults and further engagement with the autistic community is required to reach a shared understanding about whether or not these approaches should be used [21]. |
| Electro-convulsive therapy (ECT) | Not recommended | ECT is not recommended as there is evidence of a negative response to this intervention and of high risk of bias in research studies [68] Major concerns about this intervention include possible damage to brain and memory [71]. |
| Group academic and social skills training | Not recommended | There is a need for further research regarding the suitability of social skills interventions for reducing secondary effects of social skills impairments [92]. |
| Movement based interventions | Not recommended | Limited research. |
| Auditory integration training (AIT) | Not recommended | Questions have been raised regarding the potential of AIT to risk hearing loss [91]. A statement issued by American Academy of Paediatrics endorsed the lack of benefit of AIT [91]. |

GRADE = Grading of Recommendations, Assessment, Development and Evaluation

emerging evidence for the use of the PEERS programme in reducing social anxiety and loneliness [20]. Although, concerns regarding social skills interventions including PEERS includes the risk they teach camouflaging which has been associated with suicidality [71, 103]. There was also emerging evidence for social cognitive interventions [73, 83] although there is a need to explore whether these interventions are required by the autism community. There was unestablished evidence for social skills interventions [20, 77, 83, 84, 92, 101] and studies were criticised for their limited input from autistic people instead relying on parent or caregiver reports [20], Results indicate positive effects from communication interventions [20, 83, 101] but there is a need for additional robust research. Music, and dance therapies research did not demonstrate the effectiveness of this type of intervention [72, 83]. Music therapy intervention studies suggest this may have a positive impact on autistic children regarding social interaction and communication, although studies did not find significant difference in symptom severity [72]. These studies relied heavily on diagnostic measures to analyse social or behavioural differences before and after intervention [72]. Diagnostic measures are generally insensitive to change and indicate a focus on the treatment of core symptoms. There were no randomised controlled trials investigating the impact of music therapy. There is also a need for further robust research investigating the benefits of environmental, leisure and participation focussed interventions [20, 71, 73, 76, 83]. Evidence for behaviourist approaches was unestablished [71, 73, 76, 101]. Although, the

autistic community have expressed concerns regarding the use of applied behaviour analysis and further engagement is required to determine whether these approaches should be used [21]. Electro-convulsive therapy was not recommended, and autistic researchers involved in the systematic review did not feel this was an appropriate intervention for autistic people [71]. There is evidence of negative responses to this intervention and of high risk of bias in research studies [71]. Major concerns were expressed about this intervention including possible damage to brain and memory [71]. Auditory Integration Training was not recommended due to absence of evidence, and safety concerns [91]. Group academic and social skills training [84] were not recommended. Group social skills interventions were more effective for enhancing knowledge and understanding, rather than increasing specific social skills [92].

**Acceptability to the autistic community in retrieved studies.** One systematic review, exceptionally, reported including autistic individuals and families who checked results, recommendations, and acceptability of interventions [71]. Evidence for interventions aimed at the reduction of core features of autism were not recommended, Intervention studies were limited by restricted reporting of outcome measures, or use of outcome measures not validated for autistic adults.

## Discussion

Improved understanding of the relationship between individual characteristics and mental health in autistic adults is required to target interventions. Our review of studies exploring the occurrence of mental ill-health in autistic adults revealed wide variation in prevalence associated with means of diagnosis, age, co-morbidity, and country of residence. Study populations included higher proportions of male participants reflecting the historical gender imbalance in autism diagnosis.

We identified prevalence of psychiatric diagnoses in autistic adults. Attention-deficit hyperactivity disorder (2%-33%); Depression (10%-54%); and Anxiety (10%-54%) were most common. Population-based studies reflecting lifetime diagnoses identified higher prevalence than current diagnosis studies. Prevalence of mental health related diagnoses was higher in studies which used clinical samples. Most studies included smaller samples. Clinical studies may mean greater chance of clinician and service contact, raising chance of diagnosis for individuals. Diagnoses in clinical populations are also more likely to conform to identified diagnostic criteria.

This rapid review supports previous findings that age was associated with heterogeneity in prevalence of psychiatric diagnoses [1]. Autistic people over the age of 65 were more likely to report a lifetime mental health condition than autistic people aged 55–65, although this pattern was not found when examining current diagnoses [63] possibly due to additional time available to experience mental ill-health. Age related differences may be due to changes in diagnosis patterns and criteria over time, or to the reduced life expectancy of autistic people [59, 60]. Additionally, autistic people with intellectual disability are more likely to experience mental ill-health, than people with either intellectual disability or autism alone [8]. Prevalence of psychiatric diagnoses in autistic people also appears higher for people living in USA than UK [61, 62].

Prevalence may also be influenced by the lack of diagnostic tools validated for the autistic community, and which may be unable to discriminate mental ill-health from autistic features resulting in diagnostic inaccuracy or overshadowing [12, 60]. Self-reporting, and the varying ability to report internal emotional experiences, may also impact diagnostic accuracy [12].

The use of resource efficient methodologies may have reduced the number of prevalence studies revealed during this rapid review [31]. However, search strategies focussed on the

identification of research aimed at identifying the prevalence of psychiatric diagnoses in autistic adult population rather than in small purposive samples. The sampling of the population within prevalence studies is particularly important [52]. Therefore, studies which examined small purposive samples which were not compared with the wider population, were less likely to be representative of the general population.

The umbrella review revealed evidence across 31 systematic reviews relating to interventions for autistic adults, but no intervention was rated 'evidence based' and several interventions were 'not recommended.' A key issue was acceptability to the autistic community. Reviews mostly failed to consider the views of autistic adults or include autistic adults in planning or conducting the research. One systematic review, exceptionally, included autistic individuals and families who checked results, recommendations, and acceptability of interventions [71]. Research rarely focussed on the identified priorities of the autistic community which include interventions focussed on skills development and training from childhood; employment; physical health, wellbeing; mental health; and expertise, coordination, availability, and accessibility of lifespan services [48]. It should be noted that these priorities may not reflect the views autistic people who are minimally verbal.

Despite the availability of knowledge on priorities according to autistic people, the identified research mostly reflected changes to the autistic person including the development of social skills, and the reduction of 'symptoms' associated with autism or behaviours deemed undesirable. Such behavioural and psychological interventions have been criticised for aiming to remediate aspects of autism resulting in stress or harm for autistic people rather than focussing on outcomes identified as meaningful [49]. Reviewed studies incorporated a wide range of outcomes which were measured using heterogeneous tools with little discussion of their relevance to priorities identified by the autistic community. This has implications for the interpretation of the results, as measured outcomes may or may not be meaningful to autistic people and reflects the limited range of assessments validated for use with autistic adults [12].

Employment was considered in many studies and has been identified as a research priority by the autism community [48] as autistic adults are often excluded from participating in integrated competitive employment. However, studies often failed to report outcomes directly reflecting employment status following intervention [104], instead reporting outcomes such as improvement in interview skills or cognition, which may or may not support people in achieving employment. This may be due to limited follow-up after the intervention concluded. Other studies examined complex work programmes making it difficult to identify the 'effective' components.

## Implications

Further high-quality research must be designed with the autistic community focussing on their needs and priorities. Improved understanding of processes for matching individual needs and preferences with evidence-based interventions is required [3, 50]. Interventions offered to autistic people should take account of the person's preferences, needs and communication differences, and the impact these may have upon mental health [105]. Research must consider the benefits of interventions which focus on individual communication, sensory or thinking preferences. Interventions which consider adaptations to the environment must also be prioritised [50].

Staff and organisations, including health staff, adult mental health practitioners, and human resources/employment specialists, should consider what support can be offered in workplace environments [50, 71]. There are significant training needs in the workplace. This

includes training needs of autistic people and non-autistic people. Reviews particularly identified transition into employment as a key time requiring focussed attention [14].

The findings of this review suggest that practitioners and organisations who support autistic adults with their mental health should prioritise individual needs and consider focussing on approaches to building self-understanding of individual neurodevelopmental profiles before (or in conjunction with) talking therapies [71]. Ideally, practitioners should be part of a multidisciplinary team and should not only have training in approaches or therapies but have experience in working with autistic people, assessing communication support needs, and understand alternative supports and adaptations for autism [48, 49, 71]. Intervention decisions should take account of autistic people's individual preferences and needs, their day-to-day environments, neurodevelopmental and particularly communication differences, and the way sensory, communication or thinking preferences might impact on their mental health [14, 49]. Such approaches could address the current tendency to focus on people's difficulties rather than consideration of environmental supports and individual needs [49, 50].

## Inclusion of autistic people

Autistic researchers were an integral part of the research team which conducted this review and were included throughout the rapid review and umbrella review process. The team provided critique of evidence for interventions which could potentially be detrimental to autistic people, including encouraging the expression of neurotypical behaviours which is a form of masking. Papers were examined for the inclusion of autistic researchers, and for views expressed by autistic people on the acceptability of interventions.

## Limitations

Rapid review methodologies were used [34]; including date and language restrictions, limiting the number of databases that were searched, and focussing on systematic reviews of intervention studies. The research team did not have access to EMBASE which is recommended for intervention reviews where available to researchers [37]. While the review team completed a comprehensive search using recognised methods no forward or backward citation search, hand searching or follow up with authors was completed to identify missing studies. These methods and the selection of databases used in the search may have contributed to the low number of studies identified. The intervention research displayed very high heterogeneity across included studies, interventions used, and outcome measures applied. An exploratory approach to reviewing adverse effects considers only reported information, and is therefore restricted by incomplete reporting, or inadequate monitoring of adverse outcomes. Separate searches for adverse effects of interventions were not conducted and therefore results are unlikely to be comprehensive [43]. Retrieved studies did not declare that they did not include autistic researchers and research teams may therefore have included autistic researchers. Autistic researchers were integral to the research team conducting this study and their views were not recorded separately from other research team members. Autistic research team members have professional backgrounds within research, health and education and are therefore not representative of all sections of the autistic community.

## Conclusions

There is limited understanding of mental ill-health and how this can impact quality of life for autistic people despite evidence indicating increased prevalence. There is a need for diagnostic tools and outcome measures to be validated for use with this population. Future research

should fully include autistic people at every stage and focus on priorities identified by the autistic population.

## Supporting information

**S1 Checklist. PRISMA checklists.**
(DOCX)

**S1 File. Search terms.**
(DOCX)

**S2 File. Excluded citations.**
(DOCX)

**S3 File. Outcome measures.**
(DOCX)

**S4 File. Primary studies included in systematic reviews.**
(XLSX)

## Acknowledgments

**Ethical approva**l
Queen Margaret University Research Ethics Panel do not require researchers undertaking systematic review to apply for ethical approval.

## Author Contributions

**Conceptualization:** Eleanor Curnow, Marion Rutherford, Donald Maciver, Lorna Johnston, Susan Prior, Marie Boilson, Premal Shah, Natalie Jenkins, Tamsin Meff.

**Data curation:** Lorna Johnston, Natalie Jenkins.

**Formal analysis:** Marion Rutherford, Donald Maciver, Lorna Johnston, Susan Prior, Marie Boilson, Premal Shah, Tamsin Meff.

**Methodology:** Eleanor Curnow, Natalie Jenkins.

**Project administration:** Natalie Jenkins.

**Supervision:** Marion Rutherford.

**Writing – original draft:** Eleanor Curnow, Marion Rutherford, Donald Maciver.

**Writing – review & editing:** Eleanor Curnow, Marion Rutherford, Donald Maciver, Lorna Johnston, Susan Prior, Marie Boilson, Premal Shah, Natalie Jenkins, Tamsin Meff.

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
