## [Decision Letter · Decision Letter 0]

6 Nov 2022

PONE-D-22-20007Mental health in autistic adults:  a rapid systematic review of prevalence and effectiveness of interventions within a neurodiversity informed perspective

PLOS ONE

Dear Dr. Curnow,

Thank you for submitting your manuscript to PLOS ONE. After careful consideration, we feel that it has merit but does not fully meet PLOS ONE’s publication criteria as it currently stands. Therefore, we invite you to submit a revised version of the manuscript that addresses the points raised during the review process.

At this point, my decision is that the manuscript requires a major revision. You will see the comments from the two reviewers. Although you will need to consider both reviewer's suggestions, I would encourage you to particularly focus on reviewer 2's comments in your revision.

We look forward to receiving your revised manuscript.

Kind regards,

Amanda A. Webster

Academic Editor

PLOS ONE

Journal Requirements:

Reviewers' comments:

Reviewer's Responses to Questions

**Comments to the Author**

1. Is the manuscript technically sound, and do the data support the conclusions?

Reviewer #1: Yes

Reviewer #2: Partly

2. Has the statistical analysis been performed appropriately and rigorously? 

Reviewer #1: N/A

Reviewer #2: N/A

3. Have the authors made all data underlying the findings in their manuscript fully available?

Reviewer #1: Yes

Reviewer #2: Yes

4. Is the manuscript presented in an intelligible fashion and written in standard English?

Reviewer #1: Yes

Reviewer #2: Yes

5. Review Comments to the Author

Reviewer #1: Thank you for the opportunity to review this important and timely paper. The commitment to inclusive research practices is a key strength of the paper and the research team are commended on this approach. I have just a couple of comments for your consideration:

* I understand that this is a rapid review but I am left wondering what the decision making was behind the lower date range of 2011? Is there a significant event at this time that makes papers prior to this redundant? I think some justification needs to be provided for this decision.

* The inclusion of previous systematic reviews in your review was interesting - what reason drove this decision? typically previous reviews are excluded unless it is specifically a review or reviews. I wonder if focusing purely on previous reviews, elements of individual studies examining interventions were missed. I assume the rationale is that there would be too many studies, and this would then lead to the question as to whether this paper could be two separate reviews? I am not necessarily saying that this needs to be the case but clearer articulation of decision making would help here.

Reviewer #2: Thank you for the opportunity to review this manuscript. This manuscript addresses the important issue of the prevalence and interventions for mental health in autistic adults viewed through a neurodiversity paradigm. I believe this paper has potential to contribute to the literature, however I have some concerns regarding the recency and scope of the reviewed literature, feel more detail is required to justify the interpretation and conclusions, suggest a more comprehensive introduction is required to both introduce and set the context for the manuscript, and feel more detail on methods is needed to justify classifications of interventions and conclusions drawn. For ease of responding I have outlined more specific feedback by section and numbered below:

Abstract

1. Explaining why understanding mental health problems is important (e.g., to inform funding of supports, to support better quality of life) in background would be useful.

Introduction

2. The introduction is quite limited and further information to set the context for the study and the lens through which it was written would be valuable. Suggested additions are:

1. Providing a definition of mental health vs. psychiatric conditions and explaining the selection of conditions included in the review. Then using the same term throughout as I note use of both terms.

2. Overviewing prevalence of the above included conditions for neurotypical populations to provide a context for comparisons of prevalence

3. Defining the neurodiversity paradigm as well as previous research on priorities of the autistic community for research and previous research into preferred outcomes for treatments of relevance in interpreting results.

4. Defining evidence-based practice including differentiating classification based on evidence/GRADE criteria as used in the methods vs. the broader EBP framework incorporating the integration of research evidence, client priorities and preferences and practitioner expertise (cf. Sackett for example).

5. An overview of previous similar reviews in children, e.g., umbrella review of interventions e.g.,

i. Trembath, D., Varcin, K., Waddington, H., Sulek, R., Bent, C., Ashburner, J., ... & Whitehouse, A. (2022). Non-pharmacological interventions for autistic children: An umbrella review. Autism, 13623613221119368.

ii. Hossain, M. M., Khan, N., Sultana, A., Ma, P., McKyer, E. L. J., Ahmed, H. U., & Purohit, N. (2020). Prevalence of comorbid psychiatric disorders among people with autism spectrum disorder: An umbrella review of systematic reviews and meta-analyses. Psychiatry Research, 287, 112922.

6. I would like to see discussion of how this review is needed given 5.ii and what this adds (e.g., the review of interventions/autistic input?)

3. Please define and justify the use of a rapid review?

4. Alternatively, I wonder if the review of interventions may be more appropriately described as an umbrella review (see below) rather than a rapid review? See e.g.,

1. Aromataris, E., Fernandez, R., Godfrey, C. M., Holly, C., Khalil, H., & Tungpunkom, P. (2015). Summarizing systematic reviews: methodological development, conduct and reporting of an umbrella review approach. JBI Evidence Implementation, 13(3), 132-140.

5. Please provide references for assertions on page 4-5 lines 86-87 and 88-98 .

6. I would be interested to see research questions as well as aims.

Method

7. I am concerned at the relatively small number of articles found in searches, please discuss/justify selection of databases and whether likely to have captured all available studies?

8. I note the searches were conducted in November 2021, given use of systematic reviews that likely completed searches 6-12 months prior to their publication I am concerned about the recency of information, as such I suggest updating the searches and manuscript.

9. Extraction of ability level (e.g., verbal, adaptive functioning, intellectual ability) in studies is recommended where available to determine representativeness of samples.

10. Blind coding of risk of bias and reporting of Cohen’s Kappa is recommended

11. It is unclear the process by which autistic input on intervention outcomes/suitability in regard to the neurodiversity paradigm (p. 8) was completed, please describe, including inter-rater reliability/confirmation of consistency of ratings/determination of not recommended.

Results

12. For prevalence studies, where reported it would be useful to include further descriptors of participants such as whether findings differed by presence of co-occurring intellectual disability and noting if this was not described.

13. There is limited information provided in table 3 and reporting of classification of interventions to determine GRADE criteria- I am interested in the number of RCTs in each paper, the outcome measure/s, mental health conditions, and whether studies found positive, negative, or mixed results. At present there is not enough information to verify the classifications (e.g., that no intervention meets criteria as evidence-based).

14. Page 26, line 289 noted “none of the community council of autistic researchers felt this was an appropriate intervention…” as per request in method above, outlining how this process was completed and documented is needed, description of findings for each intervention would also be valuable.

Discussion

15. Gender imbalance (line 297-298) should be outlined in the introduction and reference previous research

16. Additional review in introduction is needed to then compare observed prevalence to neurotypical population (e.g., line 299-300 to discuss if these rates are higher/lower).

17. Age-related differences in lifetime prevalence likely relate to more opportunities to experience challenges with increasing age, this possibility is not discussed (see lines 309-312)

18. Discussion of elevated rates of mental health challenges in autistic people with intellectual disability vs. without should have been overviewed in the introduction (lines 313-315).

19. Discussion of limitations of current measures to detect mental health difficulties in this population (lines 316-320) would also be useful to discuss to set the scene in the introduction; overview of included measures would also be useful in the results to then critically evaluate if previous research has shown validity in this population to critically evaluate the previous findings.

20. Lines 321-313 discuss the lack of inclusion of autistic individuals in research- this is not clearly outlined in the results and should be added there and how this was determined (e.g., review of paper authorship, methods). It should also be acknowledged that research teams may include autistic individuals who chose not to self-disclose, unless it is explicitly stated that autistic researchers were not involved with community involvement statements only recently mandated and only in select journals (e.g., Autism).

21. The range of outcomes and heterogenous tools is outlined on line 338- these should be outlined in the results (e.g. outcomes and tools/measures added to Table 2).

22. Implications are not linked to previous research/findings- line 355-374 needs to link for example to autistic priorities (e.g., priority setting studies) and/or previous research findings.

23. Line 385-386 notes “other reviewers could come to different conclusions.” To address this limitation, further information about how conclusions were drawn so findings could be replicated in the method is needed, otherwise this is a serious limitation.

6. PLOS authors have the option to publish the peer review history of their article (what does this mean?). If published, this will include your full peer review and any attached files.

Reviewer #1: No

Reviewer #2: No

---

## [Author Response · Author response to Decision Letter 0]

31 Jan 2023

Reviewer Remark Author Manuscript Amendment Location

Abstract

1. Explaining why understanding mental health problems is important (e.g., to inform funding of supports, to support better quality of life) in background would be useful We have added detail regarding the importance of mental health problems to the manuscript. Autistic adults have high risk of mental ill-health and some available interventions have been associated with increased psychiatric diagnoses. Understanding prevalence of psychiatric diagnoses is important to inform the development of individualised treatment and support for autistic adults which have been identified as a research priority by the autistic community. Interventions require to be evaluated both in terms of effectiveness and regarding their acceptability to the autistic community. Page 2, lines 27-32

Introduction

2. The introduction is quite limited and further information to set the context for the study and the lens through which it was written would be valuable. Suggested additions are:

1. Providing a definition of mental health vs. psychiatric conditions and explaining the selection of conditions included in the review. Then using the same term throughout as I note use of both terms. We have enhanced the introduction as advised by the reviewer. Mental health is a state of well-being in which an individual realises his or her own abilities, can cope with the normal stresses of life, can work productively and is able to make a contribution to his or her community [15].

For the purposes of this review a psychiatric disorder is defined as a mental illness diagnosed by a mental health professional according to diagnostic criteria [17]. Relevant diagnoses were identified according to search terms and strategies described by Cochrane Common Mental Disorders [18].

 P4, 94-97, 

P5, 102-105

2. Overviewing prevalence of the above included conditions for neurotypical populations to provide a context for comparisons of prevalence We have provided additional detail on prevalence as advised by the reviewer. Worldwide prevalence of psychiatric disorders is estimated at 13%, including anxiety disorders (4.1%), depressive disorders (3.8%), bipolar disorders (0.5%), schizophrenia (0.3%), and eating disorders (0.2%) [10] . In Scotland, census data indicates that 5.4% of adults aged 16-64 years (4.6% for people aged 65+) without co-occurring intellectual disabilities and autism reported mental ill-health which had lasted or was expected to last at least 12 months [8]. P4, 82-86

3. Defining the neurodiversity paradigm as well as previous research on priorities of the autistic community for research and previous research into preferred outcomes for treatments of relevance in interpreting results. We have added information on the neurodiversity paradigm and priorities of the autistic community. The ‘neurodiversity’ movement considers autism and other neurodevelopmental conditions as neurological variation, rather than disorders requiring treatment [2, 23, 24], Therefore, autism is a difference not a deficit, which brings into question the use of interventions which seek to ‘cure, fix or normalise’ [2]. This movement has provided tools to critique research and to consider what is important in research and practice for autistic adults [16, 24, 25]. This has led to the development of research priorities which focus on the best interests of autistic people, and recognise that the inclusion of both autistic people and non-autistic people in research processes is of key importance [23]. Although, there is a need for progress as only 5% of funded autism research included autistic adults [26]. Historical research must be reviewed through a contemporary lens which considers the acceptability of terminology, interventions, supports and outcomes to the autistic community [24]. Research indicates that autistic people prioritise outcomes associated with quality of life, reduction in anxiety, depression or sleep related problems, social well-being, interpersonal relationships, and increased participation in activities of daily living, community and work [25].

 P5, 117-130

4. Defining evidence-based practice including differentiating classification based on evidence/GRADE criteria as used in the methods vs. the broader EBP framework incorporating the integration of research evidence, client priorities and preferences and practitioner expertise (cf. Sackett for example). We have added further definition of evidence based practice to the manuscript. These measures are key to evidence-based practice which requires the integration of the best available research with clinical expertise and the patient’s unique values and circumstances [27, 28]. Evidence based practice requires that health care is not only based upon the best available, valid and current evidence as defined by GRADE, but also that decisions are made by those receiving care and informed by those providing care [28, 29]. P5, 131-135

5. An overview of previous similar reviews in children, e.g., umbrella review of interventions e.g.,

i. Trembath, D., Varcin, K., Waddington, H., Sulek, R., Bent, C., Ashburner, J., ... & Whitehouse, A. (2022). Non-pharmacological interventions for autistic children: An umbrella review. Autism, 13623613221119368.

ii. Hossain, M. M., Khan, N., Sultana, A., Ma, P., McKyer, E. L. J., Ahmed, H. U., & Purohit, N. (2020). Prevalence of comorbid psychiatric disorders among people with autism spectrum disorder: An umbrella review of systematic reviews and meta-analyses. Psychiatry Research, 287, 112922. We have detailed previous umbrella reviews within the manuscript. i. A recent umbrella review found that research evidence did not support one best intervention for autism in children, and that there was a concerning lack of consideration of adverse effects of interventions [20].

ii. Whilst there has been previous consideration of prevalence of psychiatric disorders in autistic populations [11], there is a need to distinguish between adult and child populations. P5, 112-114, 

P4, 87-88

6. I would like to see discussion of how this review is needed given 5.ii and what this adds (e.g., the review of interventions/autistic input?) 5.ii refers to an umbrella review of prevalence which does not differentiate between adults and children.

We have also enhanced the argument for further review of interventions for autistic adults. Whilst there has been previous consideration of prevalence of psychiatric disorders in autistic populations [11], there is a need to distinguish between adult and child populations.

There is limited understanding of effective interventions for supporting mental health in autistic adults [2]. A recent umbrella review found that research evidence did not support one best intervention for autism in children, and that there was a concerning lack of consideration of adverse effects of interventions [20]. Previous research has focussed on children and adolescents, often evaluating interventions designed to reduce or mask behaviours associated with autism [20] but there is now recognition of the stress and detriment such interventions can create [21, 22]. The ‘neurodiversity’ movement considers autism and other neurodevelopmental conditions as neurological variation, rather than disorders requiring treatment [2, 23, 24], Therefore, autism is a difference not a deficit, which brings into question the use of interventions which seek to ‘cure, fix or normalise’ [2]. This movement has provided tools to critique research and to consider what is important in research and practice for autistic adults [16, 24, 25]. This has led to the development of research priorities which focus on the best interests of autistic people and recognise that the inclusion of both autistic people and non-autistic people in research processes is of key importance [23]. Although, there is a need for progress as only 5% of funded autism research included autistic adults [26]. Historical research must be reviewed through a contemporary lens which considers the acceptability of terminology, interventions, supports and outcomes to the autistic community [24]. Research indicates that autistic people prioritise outcomes associated with quality of life, reduction in anxiety, depression or sleep related problems, social well-being, interpersonal relationships, and increased participation in activities of daily living, community, and work [25].

 P4, 87-88,

P5, 111-130

3. Please define and justify the use of a rapid review? We have added details justifying the use of rapid review to the manuscript. The prevalence of psychiatric disorders in autistic adults will be explored through rapid review of published literature. A rapid review has been defined as “a rigorous and transparent form of knowledge synthesis which accelerates the process of conducting a traditional systematic review through streamlining or omitting a variety of methods to produce evidence for stakeholders in a resource-efficient manner” [19]. P5, 106-110

4. Alternatively, I wonder if the review of interventions may be more appropriately described as an umbrella review (see below) rather than a rapid review? See e.g.,

1. Aromataris, E., Fernandez, R., Godfrey, C. M., Holly, C., Khalil, H., & Tungpunkom, P. (2015). Summarizing systematic reviews: methodological development, conduct and reporting of an umbrella review approach. JBI Evidence Implementation, 13(3), 132-140.

The inclusion of previous systematic reviews in your review was interesting - what reason drove this decision? typically previous reviews are excluded unless it is specifically a review or reviews. I wonder if focusing purely on previous reviews, elements of individual studies examining interventions were missed. I assume the rationale is that there would be too many studies, and this would then lead to the question as to whether this paper could be two separate reviews? I am not necessarily saying that this needs to be the case but clearer articulation of decision making would help here.

 The review of systematic reviews of interventions has now been described as an umbrella review throughout the manuscript. Additionally umbrella review has been described within the manuscript:

 This umbrella review of interventions will therefore consider the results of studies not only in terms of their effectiveness, but also regarding the acceptability of the interventions to the autistic community [23].

An umbrella review facilitates a synthesis and appraisal of evidence across a broader topic area than can usually be achieved through an individual systematic review [29]. P5/6 135-138

5. Please provide references for assertions on page 4-5 lines 86-87 and 88-98 . Additional references have been added. P5 117-130

6. I would be interested to see research questions as well as aims. We have added research questions as requested by the reviewer. Research Questions:

1. How prevalent are psychiatric diagnoses in autistic adults?

2. Which factors are associated with heterogeneity of prevalence of psychiatric diagnoses in autistic adults?

3. Which interventions are effective in treating autistic adults?

4. Do available interventions meet the needs and priorities of autistic adults?

 P6 157-162

Method

7. I am concerned at the relatively small number of articles found in searches, please discuss/justify selection of databases and whether likely to have captured all available studies? Additional description regarding the selection of databases has been added. Databases were selected from resources available at Queen Margaret University which were known to publish most systematic reviews following discussion with university research librarian and with reference to published guidance [33, 34]. P7 190-192

8. I understand that this is a rapid review but I am left wondering what the decision making was behind the lower date range of 2011? Is there a significant event at this time that makes papers prior to this redundant? I think some justification needs to be provided for this decision. Citation supporting this decision has been added. Search date was restricted to 10 years as this is a valid and reliable approach for rapid reviews [34]. P7, 170-171

9. I note the searches were conducted in November 2021, given use of systematic reviews that likely completed searches 6-12 months prior to their publication I am concerned about the recency of information, as such I suggest updating the searches and manuscript. Searches have been updated to November 2022. This is reflected in the manuscript and results tables. 

10. Extraction of ability level (e.g., verbal, adaptive functioning, intellectual ability) in studies is recommended where available to determine representativeness of samples Prevalence of Intellectual Disability has been added to Prevalence study Characteristics.

Ability level, where reported, has been added to systematic reviews of interventions table. Table 2

Table 3

11. Blind coding of risk of bias and reporting of Cohen’s Kappa is recommended There was blind coding of risk of bias and this has now been made clear in the manuscript. Inter-rater agreement was assessed using Cohens Kappa. P7, 196 and 199-200

12. It is unclear the process by which autistic input on intervention outcomes/suitability in regard to the neurodiversity paradigm (p. 8) was completed, please describe, including inter-rater reliability/confirmation of consistency of ratings/determination of not recommended. We have provided additional description regarding the inclusion of autistic researchers in this study. Autistic researchers were included throughout the rapid review and umbrella review process and provided critique of evidence for interventions which could potentially be detrimental to autistic people, including encouraging the expression of neurotypical behaviours which is a form of masking. There were no incidences of disagreement between research team members regarding the classification of evidence for interventions. Papers were examined for the inclusion of autistic researchers, and for views expressed by autistic people on the acceptability of interventions. P41, 501-507

Results

13. For prevalence studies, where reported it would be useful to include further descriptors of participants such as whether findings differed by presence of co-occurring intellectual disability and noting if this was not described. Percentage of population with Intellectual Disability, where reported, has been added to Prevalence Study Characteristics Table. Table 2

14. There is limited information provided in table 3 and reporting of classification of interventions to determine GRADE criteria- I am interested in the number of RCTs in each paper, the outcome measure/s, mental health conditions, and whether studies found positive, negative, or mixed results. At present there is not enough information to verify the classifications (e.g., that no intervention meets criteria as evidence-based) Additional information (where reported) regarding ability level, outcome measures, and condition targeted has been added to table describing the characteristics of systematic reviews of interventions for autistic adults. Number of RCT’s is reported in included studies column. Table 2

15. Page 26, line 289 noted “none of the community council of autistic researchers felt this was an appropriate intervention…” as per request in method above, outlining how this process was completed and documented is needed, description of findings for each intervention would also be valuable. We have added additional information regarding the included study which employed the community council of autistic researchers. Exceptionally, one paper described the inclusion of autistic researchers within the research process [59]. This study included a community council comprising 18 people who mostly identified as autistic or were the parent of an autistic adult, and were researchers, medical or mental health professionals, authors, or advocates. This council reviewed study results and contributed to study recommendations [59]. P21, 276-280

Discussion

16. Gender imbalance (line 297-298) should be outlined in the introduction and reference previous research. We have added information regarding gender imbalance to the introduction. Estimated prevalence of autism in adults aged 16-64 years in UK is 2.9% [95% CI 2.7, 3.1] [4]. Prevalence of autism is 3.46 times higher for boys [5]. P4, 74-75

17. Additional review in introduction is needed to then compare observed prevalence to neurotypical population (e.g., line 299-300 to discuss if these rates are higher/lower) We have described overall prevalence of psychiatric disorders in the introduction. Worldwide prevalence of psychiatric disorders is estimated at 13%, including anxiety disorders (4.1%), depressive disorders (3.8%), bipolar disorders (0.5%), schizophrenia (0.3%), and eating disorders (0.2%) [10] . In Scotland, census data indicates that 5.4% of adults aged 16-64 years (4.6% for people aged 65+) without co-occurring intellectual disabilities and autism reported mental ill-health which had lasted or was expected to last at least 12 months [8]. P4, 82-86

18. Age-related differences in lifetime prevalence likely relate to more opportunities to experience challenges with increasing age, this possibility is not discussed (see lines 309-312) This possibility is now indicated in the manuscript. Autistic people over the age of 65 were more likely to report a lifetime mental health condition than autistic people aged 55-65, although this pattern was not found when examining current diagnoses [52] possibly due to additional time available to experience mental ill-health. P38/39, 425-428

19. Discussion of elevated rates of mental health challenges in autistic people with intellectual disability vs. without should have been overviewed in the introduction (lines 313-315). Needs of autistic people with intellectual disability are now overviewed in the introduction. Autistic people have a wide range of needs which vary depending on environment, and co-occurrence of intellectual or physical factors, sensory factors, co-occurring neurodevelopmental differences, intellectual disabilities, or other psychiatric diagnoses [6-8]. Autistic people, and people with intellectual disabilities have more mental and physical needs than other people [9], and research indicates that needs prevalence will be even higher for people with co-occurring autism and intellectual disability [8]. P4, 76-81

20. Discussion of limitations of current measures to detect mental health difficulties in this population (lines 316-320) would also be useful to discuss to set the scene in the introduction; overview of included measures would also be useful in the results to then critically evaluate if previous research has shown validity in this population to critically evaluate the previous findings. Additional discussion of outcome measures has been added to the introduction. The limited description of the measures prevented further investigation. Further consideration of the measurement tools used with autistic adults is also required to ensure that they are validated for this population [12].

Outcome measures were often not referenced adequately to permit investigation into their reliability or validity for this population. P4, 88-90

P21, 276-280

21. Lines 321-313 discuss the lack of inclusion of autistic individuals in research- this is not clearly outlined in the results and should be added there and how this was determined (e.g., review of paper authorship, methods). It should also be acknowledged that research teams may include autistic individuals who chose not to self-disclose, unless it is explicitly stated that autistic researchers were not involved with community involvement statements only recently mandated and only in select journals (e.g., Autism). We have added further description on the inclusion of autistic individuals and a statement regarding the absence of a statement regarding the inclusion of autistic researchers in included studies. Papers were examined for the inclusion of autistic researchers, and for views expressed by autistic people on the acceptability of interventions. 

Retrieved studies did not declare that they did not include autistic researchers and research teams may therefore have included autistic researchers. P41, 505-507, and 514-516

22. The range of outcomes and heterogenous tools is outlined on line 338- these should be outlined in the results (e.g. outcomes and tools/measures added to Table 2) Outcomes and tools/ measures have been added to intervention study characteristics table. Table 2

23. Implications are not linked to previous research/findings- line 355-374 needs to link for example to autistic priorities (e.g., priority setting studies) and/or previous research findings. Additional references have been added linking implications to previously published research/ findings. P42, 474-499

24. Line 385-386 notes “other reviewers could come to different conclusions.” To address this limitation, further information about how conclusions were drawn so findings could be replicated in the method is needed, otherwise this is a serious limitation. Additional information regarding the application of GRADE criteria has been provided. Interventions were evaluated against the stated adapted GRADE criteria to determine not only evidence of effectiveness but also evidence of negative consequences or harm. Where either of these processes indicated concern regarding the acceptability of interventions, they were categorised as not recommended. Evidence which contradicted current clinical guidelines or has been classified as not acceptable to the autistic community was not recommended.

 P9, 236-239

---

## [Decision Letter · Decision Letter 1]

20 Mar 2023

PONE-D-22-20007R1Mental health in autistic adults:  a rapid review of prevalence of psychiatric disorders and umbrella review of the effectiveness of interventions within a neurodiversity informed perspectivePLOS ONE

Dear Dr. Curnow,

Thank you for submitting your manuscript to PLOS ONE. After careful consideration, we feel that it has merit but does not fully meet PLOS ONE’s publication criteria as it currently stands. Therefore, we invite you to submit a revised version of the manuscript that addresses the points raised during the review process.

We look forward to receiving your revised manuscript.

Kind regards,

Charlotte Lennox

Academic Editor

PLOS ONE

Journal Requirements:

Reviewers' comments:

Reviewer's Responses to Questions

**Comments to the Author**

1. If the authors have adequately addressed your comments raised in a previous round of review and you feel that this manuscript is now acceptable for publication, you may indicate that here to bypass the “Comments to the Author” section, enter your conflict of interest statement in the “Confidential to Editor” section, and submit your "Accept" recommendation.

Reviewer #1: All comments have been addressed

Reviewer #2: (No Response)

2. Is the manuscript technically sound, and do the data support the conclusions?

Reviewer #1: Yes

Reviewer #2: Partly

3. Has the statistical analysis been performed appropriately and rigorously? 

Reviewer #1: N/A

Reviewer #2: N/A

4. Have the authors made all data underlying the findings in their manuscript fully available?

Reviewer #1: Yes

Reviewer #2: Yes

5. Is the manuscript presented in an intelligible fashion and written in standard English?

Reviewer #1: Yes

Reviewer #2: Yes

6. Review Comments to the Author

Reviewer #1: Thank you for re-submitting this manuscript. All changes have been addressed and this strengthens the final paper - well done to the authorship team.

Reviewer #2: Thank you for the opportunity to review this revised manuscript. This manuscript addresses the important issue of the prevalence and interventions for mental health in autistic adults viewed through a neurodiversity paradigm. The authors have addressed most of my feedback from the previous revision including updating the search which is noted as a significant revision, however a small number of queries remain particularly in regards to how the review was conducted through a neurodiversity lens and what input/feedback autistic team members provided on papers and how this influenced categorisations of levels of evidence. For ease of responding I have outlined specific feedback by section and numbered below:

Introduction

1. The definition of a rapid review does not quite fit where it is placed in text (p. 5, line 105-109), I suggest moving it to p. 6 with discussion of the umbrella review component, paraphrasing instead of directly quoting, and operationalising what about this review specifically made it a rapid review as opposed to a systematic review. Some of this information is included later in the methods but I felt this would be valuable together to set the scene for the paper.

2. GRADE acronym is not defined or operationalised on p. 5 which I suggest unpacking.

Method

3. I continue to be concerned at the relatively small number of articles found in searches, please discuss/justify selection of databases and/or highlight this more strongly as a limitation in the discussion.

4. P. 9 it is noted that “not only evidence of effectiveness but also evidence of negative consequences or harm.” Table 1 outlines how evidence of effectiveness was assessed but how was evidence of negative consequences or harm evaluated?

5. It remains unclear the process by which autistic input on intervention outcomes/suitability in regard to the neurodiversity paradigm (p. 9) was completed, please describe, including inter-rater reliability/confirmation of consistency of ratings/determination of not recommended.

6. While Cohen’s Kappa (0.70) is acceptable it indicates there were a number of discrepancies, how were these handled?

Results

7. Table 3 has been expanded to provide additional information, however there remains limited information provided about the study findings (effectiveness: positive, negative and/or mixed results AND negative consequences- not reported) to determine GRADE criteria. At present there is not enough information to verify the classifications (e.g., that no intervention meets criteria as evidence-based).

8. I continue to be interested in the input and outcomes of the community council of autistic researchers for each intervention which based on the methods was used to inform GRADE classifications but does not seem to have been reported in the results? This would be valuable information to include.

Discussion

9. Page 39, line 446-447, “A key issue was acceptability to the autistic community.” As per point 8 above these findings should be outlined in the results section.

10. Similarly, p. 41, “Autistic researchers….provided critique of evidence for interventions which could potentially be detrimental to autistic people, including encouraging the expression of neurotypical behaviours…”- this critique should be outlined in the results section for each study (e.g., add to Table 3 and summarise findings across studies).

7. PLOS authors have the option to publish the peer review history of their article (what does this mean?). If published, this will include your full peer review and any attached files.

Reviewer #1: **Yes: **Charlotte Brownlow

Reviewer #2: No

---

## [Author Response · Author response to Decision Letter 1]

30 Mar 2023

Response to Reviewers

The authors wish to thank the reviewers for their in-depth consideration of this manuscript. All points raised by reviewers are addressed below including details of manuscript amendments.

Introduction

1. The definition of a rapid review does not quite fit where it is placed in text (p. 5, line 105-109), I suggest moving it to p. 6 with discussion of the umbrella review component, paraphrasing instead of directly quoting, and operationalising what about this review specifically made it a rapid review as opposed to a systematic review. Some of this information is included later in the methods but I felt this would be valuable together to set the scene for the paper.

The authors have moved the statement defining rapid review as advised by reviewer:

P5, line 127-131 The prevalence of psychiatric disorders in autistic adults will be explored through rapid review of published literature. This knowledge synthesis will be rigorous and transparent but will be accelerated through the use of resource-efficient methods including limiting the number of databases which will be searched for evidence. Handsearching, and forward and backward citation searches will also not be undertaken[30].

2. GRADE acronym is not defined or operationalised on p. 5 which I suggest unpacking.

The authors have amended the text as follows:

P5, line 120-123 Evidence based practice requires that health care is not only based upon the best available, valid, and current evidence as defined by GRADE[28] (Grading of Recommendations, Assessment, Development and Evaluation)[28], but also that decisions are made by those receiving care and informed by those providing care[27, 29].

Method

3. I continue to be concerned at the relatively small number of articles found in searches, please discuss/justify selection of databases and/or highlight this more strongly as a limitation in the discussion.

The authors have highlighted limitations in databases used within the limitations of the study:

P51, line 522-528 Rapid review methodologies were used[33]; including date and language restrictions, limiting the number of databases that were searched, and focussing on systematic reviews of intervention studies. The research team did not have access to EMBASE which is recommended for intervention reviews where available to researchers [38]. While the review team completed a comprehensive search using recognised methods no forward or backward citation search, hand searching or follow up with authors was completed to identify missing studies. These methods may have reduced the number of studies identified.

4. P. 9 it is noted that “not only evidence of effectiveness but also evidence of negative consequences or harm.” Table 1 outlines how evidence of effectiveness was assessed but how was evidence of negative consequences or harm evaluated?

The authors have provided additional information regarding the evaluation of negative consequences or harm:

P9, line 229-240

Interventions were evaluated against the stated adapted GRADE criteria to determine not only evidence of effectiveness, but also evidence of negative consequences or harm. This involved consideration of reported benefit and adverse effects for each intervention type. Interventions which focussed on the reduction of core features of autism are associated with harmful consequences and contradict current clinical guidelines so were rated as not recommended[42]. All members of the research team which included autistic researchers, were involved in this process. Arising disagreements were resolved through team discussion with reference to the principles of the neurodiversity paradigm[43] and expressed priorities and concerns of autistic adults[14, 44, 45]. Where either of these processes indicated concern regarding the acceptability of an intervention, it was categorised as not recommended. Interventions which contradicted current clinical guidelines were also not recommended.

5. It remains unclear the process by which autistic input on intervention outcomes/suitability in regard to the neurodiversity paradigm (p. 9) was completed, please describe, including inter-rater reliability/confirmation of consistency of ratings/determination of not recommended.

• The authors have enhanced the description of negative consequences or harm to clarify the connection between the neurodiversity paradigm, neurodiversity affirming practice and consequences of intervention (see response above). 

• There is no calculation of inter-rater reliability as this process was applied to interventions across different systematic review studies. 

• Ratings were agreed by the entire team, following discussion as described on P9, line 234-237

All members of the research team which included autistic researchers, were involved in this process. Arising disagreements were resolved through team discussion with reference to the principles of the neurodiversity paradigm[43] and expressed priorities and concerns of autistic adults[14, 44, 45].

6. While Cohen’s Kappa (0.70) is acceptable it indicates there were a number of discrepancies, how were these handled?

P7, line 191-192

Disagreements were to be resolved through discussion and reference to a third party (MR) but this was not required. 

Results

7. Table 3 has been expanded to provide additional information, however there remains limited information provided about the study findings (effectiveness: positive, negative and/or mixed results AND negative consequences- not reported) to determine GRADE criteria. At present there is not enough information to verify the classifications (e.g., that no intervention meets criteria as evidence-based).

P23, Table 3: Additional columns have been added to this table to show (1) Effectiveness (+, - or Mixed), and (2) Negative Consequences of Interventions.

8. I continue to be interested in the input and outcomes of the community council of autistic researchers for each intervention which based on the methods was used to inform GRADE classifications but does not seem to have been reported in the results? This would be valuable information to include.

The authors wish to state for clarification, a community council of autistic researchers is not included in methods of this study. The community of council of autistic researchers referred to in the text within the results section were part of one of the reported studies and not this review. However, the research team responsible for this study included autistic researchers. Table 4 includes the GRADE rating, together with key issues relating to the acceptability of interventions to autistic adults agreed following team discussion. 

Discussion

9. Page 39, line 446-447, “A key issue was acceptability to the autistic community.” As per point 8 above these findings should be outlined in the results section.

Table 4 Evidence for Interventions

Column 3 has been extended to include additional information regarding the acceptability of interventions to autistic adults.

10. Similarly, p. 41, “Autistic researchers….provided critique of evidence for interventions which could potentially be detrimental to autistic people, including encouraging the expression of neurotypical behaviours…”- this critique should be outlined in the results section for each study (e.g., add to Table 3 and summarise findings across studies).

Table 4 Evidence for Interventions

Column 3 of this table has been amended to include information considered by the research team when making decisions regarding the acceptability of interventions to autistic adults.

---

## [Editor Report · Decision Letter 2]

18 Apr 2023

PONE-D-22-20007R2Mental health in autistic adults:  a rapid review of prevalence of psychiatric disorders and umbrella review of the effectiveness of interventions within a neurodiversity informed perspectivePLOS ONE

Dear Dr. Curnow,

Thank you for submitting your manuscript to PLOS ONE. After careful consideration, we feel that it has merit but does not fully meet PLOS ONE’s publication criteria as it currently stands. Therefore, we invite you to submit a revised version of the manuscript that addresses the points raised during the review process. One of the reviewers has requested some additional minor changes, please see below.   Please submit your revised manuscript by Jun 02 2023 11:59PM. If you will need more time than this to complete your revisions, please reply to this message or contact the journal office at plosone@plos.org. Please include the following items when submitting your revised manuscript:A rebuttal letter that responds to each point raised by the academic editor and reviewer(s). You should upload this letter as a separate file labeled 'Response to Reviewers'.A marked-up copy of your manuscript that highlights changes made to the original version. You should upload this as a separate file labeled 'Revised Manuscript with Track Changes'.An unmarked version of your revised paper without tracked changes. You should upload this as a separate file labeled 'Manuscript'.If applicable, we recommend that you deposit your laboratory protocols in protocols.io to enhance the reproducibility of your results. Protocols.io assigns your protocol its own identifier (DOI) so that it can be cited independently in the future. For instructions see: https://journals.plos.org/plosone/s/submission-guidelines#loc-laboratory-protocols. Additionally, PLOS ONE offers an option for publishing peer-reviewed Lab Protocol articles, which describe protocols hosted on protocols.io. Read more information on sharing protocols at https://plos.org/protocols?utm_medium=editorial-email&utm_source=authorletters&utm_campaign=protocols.

We look forward to receiving your revised manuscript.

Kind regards,

Charlotte Lennox

Academic Editor

PLOS ONE

Journal Requirements:

Reviewers' comments:

Introduction

1. The definition of a rapid review does not quite fit where it is placed in text (p. 5, line 105-109), I suggest moving it to p. 6 with discussion of the umbrella review component, paraphrasing instead of directly quoting, and operationalising what about this review specifically made it a rapid review as opposed to a systematic review. Some of this information is included later in the methods but I felt this would be valuable together to set the scene for the paper.

2. GRADE acronym is not defined or operationalised on p. 5 which I suggest unpacking.

Method

3. I continue to be concerned at the relatively small number of articles found in searches, please discuss/justify selection of databases and/or highlight this more strongly as a limitation in the discussion.

4. P. 9 it is noted that “not only evidence of effectiveness but also evidence of negative consequences or harm.” Table 1 outlines how evidence of effectiveness was assessed but how was evidence of negative consequences or harm evaluated?

5. It remains unclear the process by which autistic input on intervention outcomes/suitability in regard to the neurodiversity paradigm (p. 9) was completed, please describe, including inter-rater reliability/confirmation of consistency of ratings/determination of not recommended.

6. While Cohen’s Kappa (0.70) is acceptable it indicates there were a number of discrepancies, how were these handled?

Results

7. Table 3 has been expanded to provide additional information, however there remains limited information provided about the study findings (effectiveness: positive, negative and/or mixed results AND negative consequences- not reported) to determine GRADE criteria. At present there is not enough information to verify the classifications (e.g., that no intervention meets criteria as evidence-based).

8. I continue to be interested in the input and outcomes of the community council of autistic researchers for each intervention which based on the methods was used to inform GRADE classifications but does not seem to have been reported in the results? This would be valuable information to include.

Discussion

9. Page 39, line 446-447, “A key issue was acceptability to the autistic community.” As per point 8 above these findings should be outlined in the results section.

10. Similarly, p. 41, “Autistic researchers….provided critique of evidence for interventions which could potentially be detrimental to autistic people, including encouraging the expression of neurotypical behaviours…”- this critique should be outlined in the results section for each study (e.g., add to Table 3 and summarise findings across studies).

---

## [Author Response · Author response to Decision Letter 2]

28 Apr 2023

Mental health in autistic adults: a rapid review of prevalence of psychiatric disorders and umbrella review of the effectiveness of interventions within a neurodiversity informed perspective

The authors wish to thank the reviewers and the editor for their consideration of this study. The authors have extensively revised the manuscript in response to their suggestions. Please find our responses to your suggested revisions below. We look forward to hearing your decision regarding publication of this manuscript in due course.

Reviewers' comments:

Introduction

1. The definition of a rapid review does not quite fit where it is placed in text (p. 5, line 105-109), I suggest moving it to p. 6 with discussion of the umbrella review component, paraphrasing instead of directly quoting, and operationalising what about this review specifically made it a rapid review as opposed to a systematic review. Some of this information is included later in the methods but I felt this would be valuable together to set the scene for the paper.

Author Response:

P6, line128-134 We previously moved definition of rapid review to its current location prior to the description of the umbrella review as requested. We have added further details regarding the rapid review methods used within the study.

The prevalence of psychiatric disorders in autistic adults will be explored through rapid review of published literature. This knowledge synthesis will be rigorous and transparent but will be accelerated by resource-efficient methods including limiting the number of databases which will be searched for evidence. Handsearching, and forward and backward citation searches will also not be undertaken[30]. Grey literature, and literature not published in English will not be considered. Article screening will be reviewed by two authors in 20% of publications.

2. GRADE acronym is not defined or operationalised on p. 5 which I suggest unpacking.

Author Response:

P5, line 120-124

Evidence based practice requires that health care is not only based upon the best available, valid, and current evidence as defined by GRADE (Grading of Recommendations, Assessment, Development and Evaluation)[28], but also that decisions are made by those receiving care and informed by those providing care[27, 29]. Strong GRADE evidence indicates all or almost all people would choose that intervention[30]. 

Method

3. I continue to be concerned at the relatively small number of articles found in searches, please discuss/justify selection of databases and/or highlight this more strongly as a limitation in the discussion.

Author response:

P7, line 186-192

Databases were selected from available resources following current guidance[37], and through discussion with the university research librarian. CENTRAL, MEDLINE and Embase (if access to Embase is available to the review team) are recommended for systematic reviews[37-39] . Embase was not available to the research team. Cochrane Database of Systematic Reviews was included as a major repository of systematic reviews[32]. Trials searches of JBI Database of Systematic Reviews and Implementation Reports did not reveal any additional relevant citations and was therefore excluded. 

P52, line 538-544 Additionally we have provided additional information in the limitations section highlighting the possible impact of rapid review methods including database limitations on the number of identified studies.

Limitations

Rapid review methodologies were used[34]; including date and language restrictions, limiting the number of databases that were searched, and focussing on systematic reviews of intervention studies. The research team did not have access to EMBASE which is recommended for intervention reviews where available to researchers[37]. While the review team completed a comprehensive search using recognised methods no forward or backward citation search, hand searching or follow up with authors was completed to identify missing studies. These methods may have contributed to the low number of studies identified. 

4. P. 9 it is noted that “not only evidence of effectiveness but also evidence of negative consequences or harm.” Table 1 outlines how evidence of effectiveness was assessed but how was evidence of negative consequences or harm evaluated?

Author Response

P9, line 238-241 The authors have added additional information regarding their approach to the consideration of adverse outcomes.

An exploratory approach was used to review adverse outcomes identified during the conduct of the review. This opportunistic approach considers only the reported adverse effects or outcomes that may be associated with the interventions being investigated[43].

P52, line 550-554 The authors have also added the following limitations in this regard.

An exploratory approach to reviewing adverse effects considers only reported information, and is therefore restricted by incomplete reporting, or inadequate monitoring of adverse outcomes. Separate searches for adverse effects of interventions were not conducted and therefore results are unlikely to be comprehensive[42].

5. It remains unclear the process by which autistic input on intervention outcomes/suitability in regard to the neurodiversity paradigm (p. 9) was completed, please describe, including inter-rater reliability/confirmation of consistency of ratings/determination of not recommended.

Author Response

P9, line 241-249 We have simplified the description of this process and hope that it is now clearer to the reader.

Interventions which focussed on the reduction of core features of autism are associated with harmful consequences and contradict current clinical guidelines so were rated as not recommended [44]. Core features include qualitative differences and impairments in reciprocal social interaction and social communication, restricted interests and activities, and rigid and repetitive behaviours[45]. Interventions which contradicted current clinical guidelines were also not recommended. All members of the research team which included autistic researchers, were involved in this process. Arising disagreements were resolved through team discussion. Inter-rater reliability was not recorded for this process. 

6. While Cohen’s Kappa (0.70) is acceptable it indicates there were a number of discrepancies, how were these handled?

Author Response

P8, Line 198-199

Disagreements were resolved through discussion, and reference to a third party (MR) was not required.

Results

7. Table 3 has been expanded to provide additional information, however there remains limited information provided about the study findings (effectiveness: positive, negative and/or mixed results AND negative consequences- not reported) to determine GRADE criteria. At present there is not enough information to verify the classifications (e.g., that no intervention meets criteria as evidence-based).

Author Response.

Negative consequences are reported in Table 3 Column 9. Additional information regarding the effect sizes of meta-analysis or quantitative synthesis conducted within the retrieved systematic reviews are reported in Table 3 Column 8. 

Additionally we have included supplementary data describing overlap of primary studies included in the retrieved systematic reviews.

S3 File Primary Studies included in Systematic Reviews

8. I continue to be interested in the input and outcomes of the community council of autistic researchers for each intervention which based on the methods was used to inform GRADE classifications but does not seem to have been reported in the results? This would be valuable information to include.

Author response.

As stated in the previous rebuttal, this review did not include a community council of autistic researchers, although autistic researchers were integrated into the research team and included in all aspects of the review. Community council of autistic researchers is not included in the methods.

Discussion

9. Page 39, line 446-447, “A key issue was acceptability to the autistic community.” As per point 8 above these findings should be outlined in the results section.

Author Response.

P52, line 433-437 

One systematic review, exceptionally, reported including autistic individuals and families who checked results, recommendations, and acceptability of interventions[64]. Evidence for interventions aimed at the reduction of core features of autism were not recommended, Intervention studies were limited by not restricted reporting of outcome measures, or use of outcome measures not validated for autistic adults. 

10. Similarly, p. 41, “Autistic researchers….provided critique of evidence for interventions which could potentially be detrimental to autistic people, including encouraging the expression of neurotypical behaviours…”- this critique should be outlined in the results section for each study (e.g., add to Table 3 and summarise findings across studies).

Author Response

Table 4 Column 3 includes details regarding the acceptability of interventions to the autistic community. As most of the retrieved studies included a number of different types of interventions the authors have included this information in Table 4 which provided a summary of interventions.

---

## [Editor Report · Decision Letter 3]

15 May 2023

PONE-D-22-20007R3Mental health in autistic adults:  a rapid review of prevalence of psychiatric disorders and umbrella review of the effectiveness of interventions within a neurodiversity informed perspectivePLOS ONE

Dear Dr. Curnow,

Thank you for submitting your manuscript to PLOS ONE. After careful consideration, we feel that it has merit but does not fully meet PLOS ONE’s publication criteria as it currently stands. Therefore, we invite you to submit a revised version of the manuscript that addresses the points raised during the review process.

I'd like to thank the authors for addressing the previous reviewers comments.  One reviewer is now happy to accept the manuscript, however one reviewer has asked for a few minor changes.  Some are very minor, having looked at the comments (all detailed below), I would appreciate if the authors could address the following: 

Method

3. I continue to be concerned at the relatively small number of articles found in searches, please discuss/justify selection of databases and/or highlight this more strongly as a limitation in the discussion.

5. It remains unclear the process by which autistic input on intervention outcomes/suitability in regard to the neurodiversity paradigm (p. 9) was completed, please describe, including inter-rater reliability/confirmation of consistency of ratings/determination of not recommended.

6. While Cohen’s Kappa (0.70) is acceptable it indicates there were a number of discrepancies, how were these handled?

Results

7. Table 3 has been expanded to provide additional information, however there remains limited information provided about the study findings (effectiveness: positive, negative and/or mixed results AND negative consequences- not reported) to determine GRADE criteria. At present there is not enough information to verify the classifications (e.g., that no intervention meets criteria as evidence-based).

We look forward to receiving your revised manuscript.

Kind regards,

Charlotte Lennox

Academic Editor

PLOS ONE

Journal Requirements:

Reviewers' comments:

Introduction

1. The definition of a rapid review does not quite fit where it is placed in text (p. 5, line 105-109), I suggest moving it to p. 6 with discussion of the umbrella review component, paraphrasing instead of directly quoting, and operationalising what about this review specifically made it a rapid review as opposed to a systematic review. Some of this information is included later in the methods but I felt this would be valuable together to set the scene for the paper.

2. GRADE acronym is not defined or operationalised on p. 5 which I suggest unpacking.

Method

3. I continue to be concerned at the relatively small number of articles found in searches, please discuss/justify selection of databases and/or highlight this more strongly as a limitation in the discussion.

4. P. 9 it is noted that “not only evidence of effectiveness but also evidence of negative consequences or harm.” Table 1 outlines how evidence of effectiveness was assessed but how was evidence of negative consequences or harm evaluated?

5. It remains unclear the process by which autistic input on intervention outcomes/suitability in regard to the neurodiversity paradigm (p. 9) was completed, please describe, including inter-rater reliability/confirmation of consistency of ratings/determination of not recommended.

6. While Cohen’s Kappa (0.70) is acceptable it indicates there were a number of discrepancies, how were these handled?

Results

7. Table 3 has been expanded to provide additional information, however there remains limited information provided about the study findings (effectiveness: positive, negative and/or mixed results AND negative consequences- not reported) to determine GRADE criteria. At present there is not enough information to verify the classifications (e.g., that no intervention meets criteria as evidence-based).

8. I continue to be interested in the input and outcomes of the community council of autistic researchers for each intervention which based on the methods was used to inform GRADE classifications but does not seem to have been reported in the results? This would be valuable information to include.

Discussion

9. Page 39, line 446-447, “A key issue was acceptability to the autistic community.” As per point 8 above these findings should be outlined in the results section.

10. Similarly, p. 41, “Autistic researchers….provided critique of evidence for interventions which could potentially be detrimental to autistic people, including encouraging the expression of neurotypical behaviours…”- this critique should be outlined in the results section for each study (e.g., add to Table 3 and summarise findings across studies).

---

## [Author Response · Author response to Decision Letter 3]

2 Jun 2023

The authors wish to thank the reviewers for their time and careful consideration of the content of this manuscript. We appreciate their work and have amended the manuscript in response to their comments. We hope we have understood their requirements and responded appropriately. We have described the changes we have made to the manuscript below.

Method

3. I continue to be concerned at the relatively small number of articles found in searches, please discuss/justify selection of databases and/or highlight this more strongly as a limitation in the discussion.

Author Response:

We have discussed the selection of databases (p7, line 186-191). We have provided additional reference to database selection as a concern within the limitations section of the manuscript.

P61, line 589-596

Rapid review methodologies were used [34]; including date and language restrictions, limiting the number of databases that were searched, and focussing on systematic reviews of intervention studies. The research team did not have access to EMBASE which is recommended for intervention reviews where available to researchers [37]. While the review team completed a comprehensive search using recognised methods no forward or backward citation search, hand searching or follow up with authors was completed to identify missing studies. These methods and the selection of databases used in the search may have contributed to the low number of studies identified. 

5. It remains unclear the process by which autistic input on intervention outcomes/suitability in regard to the neurodiversity paradigm (p. 9) was completed, please describe, including inter-rater reliability/confirmation of consistency of ratings/determination of not recommended.

Author Response:

We have provided additional detail regarding the principles by which we considered the acceptability of interventions, in addition to criteria included in Table 1.

P10, line 241-254

In considering negative consequences or harm associated with interventions we included criteria adapted from clinical guidelines and neurodiversity affirming practice. Specifically, we did not recommend:

• Interventions which focussed on the reduction of core features of autism are associated with harmful consequences and contradict current clinical guidelines [44, 45]. Core features include qualitative differences and impairments in reciprocal social interaction and social communication, restricted interests and activities, and rigid and repetitive behaviours [46]. 

• Interventions which contradicted current clinical guidelines [44, 46]. 

• Interventions associated with adverse events or adverse outcomes [43].

• Interventions which attempt to ‘cure, fix or normalise’ autistic people [2, 47] due to their negative impact upon quality of life [29].

• Interventions which target outcomes contradictory to the identified priorities of the autistic community [14, 24, 48-50].

We have provided additional information regarding the role of autistic and non-autistic researchers within the research team.

P10, line 255-266

The research team was made up of autistic and non-autistic professionals within speech and language therapy, psychology, psychiatry, occupational therapy, and teaching fields. Members of the team had research experience, and experience working with autistic people in clinical and education settings. As integrated members of the research team, autistic researchers contributed to the planning and design of this research study, and decision-making related to study outcomes alongside non-autistic colleagues. All team members held professional roles and contributed expertise to the study thus possibly reducing issues associated with power hierarchy sometimes found in autism research [51]. Arising disagreements concerning the classification of evidence were resolved through team discussion with reference to research recommendation classification (Table 1) and criteria regarding negative consequences or harm listed above. Inter-rater reliability was not recorded for this process. 

Additionally, we have provided reference to this within the study limitations.

P62, line 603-606

Autistic researchers were integral to the research team this study and their views were not recorded separately from other research team members. Autistic research team members have professional backgrounds within research, health and education and are therefore not representative of all sections of the autistic community.

6. While Cohen’s Kappa (0.70) is acceptable it indicates there were a number of discrepancies, how were these handled?

Author Response:

We have provided additional detail regarding the process for determining the outcome of arising discrepancies.

P10, line 263-266

Arising disagreements regarding the classification of evidence were resolved through team discussion with reference to research recommendation classification (Table 1) and criteria regarding negative consequences or harm listed above until agreement was achieved. Inter-rater reliability was not recorded for this process.

Results

7. Table 3 has been expanded to provide additional information, however, there remains limited information provided about the study findings (effectiveness: positive, negative and/or mixed results AND negative consequences- not reported) to determine GRADE criteria. At present there is not enough information to verify the classifications (e.g., that no intervention meets criteria as evidence-based).

Author Response:

Table 3 has been enhanced to include additional information on effectiveness of interventions, and negative consequences reported within the systematic reviews.

P50, line 383-387

Column 3 of Table 4 outlines factors which may impact the acceptability of interventions to autistic adults including research limitations, indications of adverse effects, adverse outcomes, or priorities contradicting those identified by the autism community. Overall, results indicate a need for further robust research.

We have also expanded the information included in Table 4 to provide additional detail regarding the effectiveness and acceptability of interventions.

These details are expanded throughout the sections describing the outcomes by type of intervention.

P56, line 397-483

Pharmacological interventions

Five reviews [81, 88, 89, 98, 100] considered 139 studies evaluating pharmacological intervention for autistic individuals. One review was high quality (Table 3). Managing behaviours with medication as a first line of intervention or using medication including SSRIs (Selective Serotonin Reuptake Inhibitors) or Oxytocin for core features of Autism is not recommended (Table 4) [46, 100]. However, there was emerging evidence for use of medication as a last line of intervention. Oxytocin may offer some benefit but did not affect global clinical status [88]. Risperidone may be useful in the management of repetitive, aggressive, and self-injurious behaviour [81], although side-effects are problematic [98]. There was limited evidence to support the use of opioid antagonists to reduce self-injury in autistic adults [89]. However, fluoxetine or fluvoxamine may be useful in the management of repetitive and obsessive-compulsive behaviour and anxiety where other interventions are not available or possible due to the individual’s level of distress or aggression [98]. Overall, there is a need for future research to consider the acceptability of pharmacological interventions including further investigation of side-effects.

Employment focused interventions 

Nine reviews of evidence for employment focussed interventions considered 100 unique publications [78, 80, 82, 83, 85-87, 90, 97]. None of the reviews were high quality (Table 3). Reviews revealed emerging evidence that supported employment including Individual Placement Support (IPS) and Project Search, yields positive outcomes for autistic people [78, 80, 82, 86, 87, 90, 97]. Notably, autistic adults, undertaking Project SEARCH with autism support were eleven times more likely to achieve employment than those attending special education [87]. However methodological concerns mean this result must be interpreted with caution as studies did not include comparable control groups or consider participant attrition [78]. Evidence for technology-supported interventions such as virtual reality training was unestablished as the relationship to paid employment was not confirmed [85]. Employment related social skills training research often focussed on alternative outcomes to employment status, such as interview skills performance, and therefore the evidence for such an approach is unestablished. Sheltered workshops were not recommended as they were not associated with supporting autistic people into employment but could provide other benefits. Further research is required to consider the impact of employment focussed interventions not only on employment status and wage, but also on quality of life [24].

Psychological therapies

There were 7 reviews of psychological therapies including 215 studies [74, 75, 79, 93-95, 99]. Only one review was of high quality (Table 3). The reviews revealed emerging evidence (Table 4) for the use of mindfulness for the reduction of self-reported depression symptoms in autistic adults without intellectual disability [71, 79, 84]. Studies provided emerging evidence for use of Cognitive remediation therapy to improve cognitive function, but small sample sizes and limited follow-up made it difficult to determine meaningful impact or maintenance of any benefit in the longer term [75]. 

There was unestablished evidence for the use of cognitive behavioural therapy (CBT), although small positive clinical effects on self-reported outcomes were observed [71, 99]. Within nine systematic reviews, which included CBT studies, 11 different types of CBT were described and included CBT combined with other interventions including behavioural techniques, mindfulness, and psychoeducation [93, 94].These major variations in the intervention provided meant it was not possible to conclude this intervention was effective. Additionally, there were expressed concerns regarding CBT which are outlined in table 4 and which should be considered in future research.

There was unestablished evidence for family therapy due to limited quality research [95] although non-randomised intervention studies suggest there may be improved knowledge and understanding of core disorder (ASD), and coping styles post-intervention [95]. Acceptance and Commitment Therapy was not recommended due to limited research and insufficient rigour [74] to suggest ACT is effective in the management of psychological distress for individuals with ID [74].

Mixed interventions and approaches

Twelve systematic reviews considered 300 studies within 11 sub-categories of intervention identified [20, 71-73, 76, 77, 83, 84, 91, 92, 96, 101]. Two reviews were rated as high quality (Table 3). Evidence for most of interventions in this grouping was unestablished or not recommended (Table 4). However, there was emerging evidence for the use of the PEERS programme in reducing social anxiety and loneliness [20]. Although, concerns regarding social skills interventions including PEERS includes the risk they teach camouflaging which has been associated with suicidality [71, 103]. There was also emerging evidence for social cognitive interventions [73, 83] although there is a need to explore whether these interventions are required by the autism community. There was unestablished evidence for social skills interventions [20, 77, 83, 84, 92, 101] and studies were criticised for their limited input from autistic people instead relying on parent or caregiver reports [20], Results indicate positive effects from communication interventions [20, 83, 101] but there is a need for additional robust research. Music, and dance therapies research did not demonstrate the effectiveness of this type of intervention [72, 83]. Music therapy intervention studies suggest this may have a positive impact on autistic children regarding social interaction and communication, although studies did not find significant difference in symptom severity [72]. These studies relied heavily on diagnostic measures to analyse social or behavioural differences before and after intervention [72]. Diagnostic measures are generally insensitive to change and indicate a focus on the treatment of core symptoms. There were no randomised controlled trials investigating the impact of music therapy. There is also a need for further robust research investigating the benefits of environmental, leisure and participation focussed interventions [20, 71, 73, 76, 83]. Evidence for behaviourist approaches was unestablished [71, 73, 76, 101]. Although, the autistic community have expressed concerns regarding the use of applied behaviour analysis and further engagement is required to determine whether these approaches should be used [21]. Electro-convulsive therapy was not recommended, and autistic researchers involved in the systematic review did not feel this was an appropriate intervention for autistic people [71]. There is evidence of negative responses to this intervention and of high risk of bias in research studies [71]. Major concerns were expressed about this intervention including possible damage to brain and memory [71]. Auditory Integration Training was not recommended due to absence of evidence, and safety concerns [91]. Group academic and social skills training [84] were not recommended. Group social skills interventions were more effective for enhancing knowledge and understanding, rather than increasing specific social skills [92].

---

## [Editor Report · Decision Letter 4]

26 Jun 2023

Mental health in autistic adults:  a rapid review of prevalence of psychiatric disorders and umbrella review of the effectiveness of interventions within a neurodiversity informed perspective

PONE-D-22-20007R4

Dear Dr. Eleanor Curnow,

We’re pleased to inform you that your manuscript has been judged scientifically suitable for publication and will be formally accepted for publication once it meets all outstanding technical requirements.

Kind regards,

Weihua YUE, M.D.

Academic Editor

PLOS ONE
---

## [Editor Report · Acceptance letter]

3 Jul 2023

PONE-D-22-20007R4 

Mental health in autistic adults:  a rapid review of prevalence of psychiatric disorders and umbrella review of the effectiveness of interventions within a neurodiversity informed perspective 

Dear Dr. Curnow:

I'm pleased to inform you that your manuscript has been deemed suitable for publication in PLOS ONE. Congratulations! Your manuscript is now with our production department. 

Kind regards, 

on behalf of

Dr. Weihua YUE 

Academic Editor

PLOS ONE